# QAT-SAM: Accurate Quantization for Segment Anything Model 2

## Abstract

The Segment Anything Model 2 (SAM2) is a powerful foundation model for promptable segmentation. However, its high computational and memory costs are a major barrier to deployment on resource-constrained devices. In this paper, we present QAT-SAM, a low-bit quantization method that substantially improves robustness over prior QAT baselines at extreme bit-widths while delivering large model-size reductions. To address performance degradation arising from challenging weight and activation distributions during quantization, QAT-SAM introduces two novel contributions: Variance-Reduced Calibration (VRC), an initialization method that reduces weight statistical variance by minimizing the Frobenius norm over a small calibration batch; and Learnable Statistical Clipping (LSC), a Quantization-Aware Training (QAT) method that learns momentum-stabilized clipping factors to manage outliers in weights and activations. Comprehensive experiments demonstrate that QAT-SAM substantially closes the QAT accuracy gap to full precision at low bit-widths and significantly outperforms prior QAT baselines, particularly in the ultra-low 2-bit regime. Specifically, QAT-SAM achieves an accuracy gain of up to 9.7 ppt in J&F on the video segmentation benchmark and 7.3 ppt in mIoU for instance segmentation over the best competing QAT model, all while achieving an 8x reduction in model size compared to the BF16 baseline.

## 1 Introduction

The Segment Anything Model 2 (SAM2) has emerged as the foundational framework for unified promptable segmentation across static images and video Ravi et al. (2024). Its performance on video segmentation makes this model attractive for applications in robotics and tiny machine learning. However, taking advantage of real-time inference with this model requires high-end hardware, such as an NVIDIA A100 GPU. Furthermore, edge deployment scenarios typically impose tighter memory and storage constraints, even at the MB level in extreme cases, Han et al. (2016), which makes the use of the existing version of SAM2 prohibitive. Efficiency in large-scale models is frequently pursued through architectural redesign and knowledge distillation Zhao et al. (2023); Fu et al. (2024); Zhang et al. (2023); Bonazzi et al. (2025); Zhou et al. (2023). While these methods significantly reduce computational overhead, they often compromise the inherent zeroshot generalization of foundation models, effectively transforming them into more constrained supervised solutions.

As a versatile alternative, quantization serves as a general strategy to compress and speed up deep neural networks by utilizing fewer bits. This technique is widely adopted for optimizing models for deployment on resource-constrained hardware Gholami et al. (2022); Nagel et al. (2021). By reducing the numerical precision of weights and activations, quantization enables faster inference, lower power consumption, and smaller model sizes without necessitating fundamental changes to the model's architectural capacity Jacob et al. (2018); Banner et al. (2019).

For these reasons, the demand for post-training quantization (PTQ) methods has grown substantially in recent years, particularly due to the increasing size and complexity of foundation models like large language models (LLMs) Dettmers et al. (2022); Xiao et al. (2023); Ashkboos et al. (2024); Liu et al. (2025b); Frantar

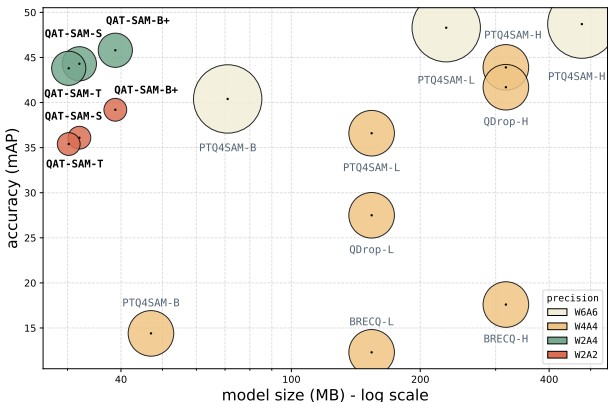

Figure 1: Comparison between QAT-SAM and Post-Training Quantization (PTQ) based on SAM Kirillov et al. (2023) [ViT-B/L/H]. QAT-SAM's W2A2 / W2A4 Hiera-based models achieve accuracy comparable to PTQ baselines (PTQ4SAM Lv et al. (2024), BRECQ Li et al. (2021), QDROP Wei et al. (2023)) running at higher precision on the original SAM (ViT-B/L/H). The figure spans two architectures (SAM ViT vs. SAM2 Hiera) and is intended as a model-size vs. accuracy reference across SAM generations, not a same-backbone PTQ-vs-QAT comparison.

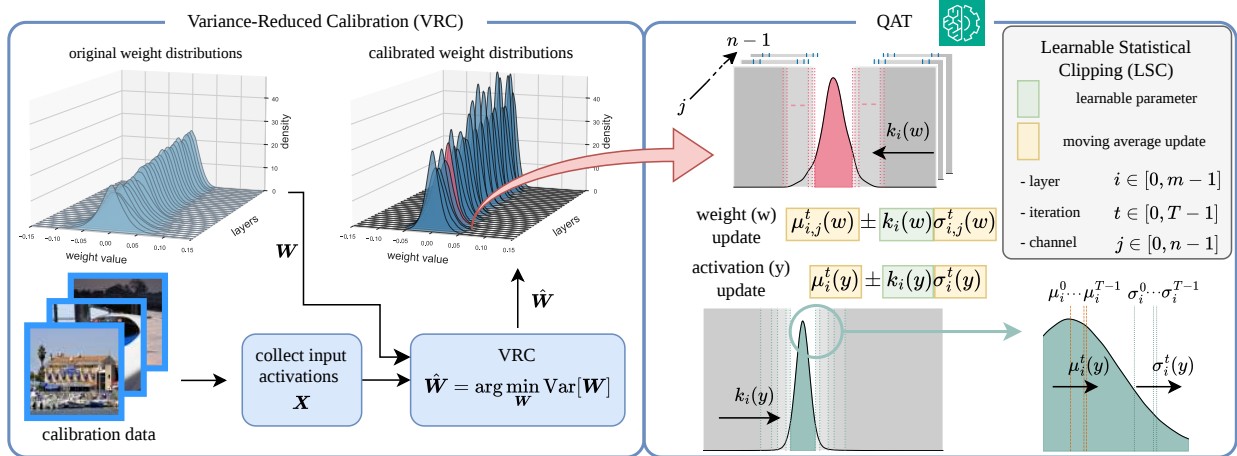

Figure 2: The QAT-SAM approach. The weight distributions of the linear layers in the image encoder are calibrated using the VRC to reduce variance. We substitute the original encoder and train the network using the LSC in a QAT pipeline.

et al. (2022); Nagel et al. (2019); Lv et al. (2024); Wu et al. (2024). These models often consist of billions of parameters, making them difficult to deploy efficiently without compression techniques. PTQ offers a practical solution by enabling model compression without requiring retraining, thus reducing computational costs and accelerating deployment Gholami et al. (2022); Nagel et al. (2020); Li et al. (2021). Quantization Aware Training (QAT) Cai et al. (2017); Zhou et al. (2016); Esser et al. (2019); Choi et al. (2018); Hubara et al. (2018), offers a more accurate solution by simulating quantization effects during the training phase, allowing the model to adapt to reduced numerical precision Jacob et al. (2018); Wu et al. (2020); Nagel et al. (2021). QAT enables the model to recover accuracy under aggressive compression, but its adoption remains limited in transformer architectures in resource-constrained scenarios due to its lack of extensive investigation and the high computational cost.

While no previous work has focused on the SAM2 architecture, recent studies have proposed PTQ algorithms for the original Segment Anything Model Kirillov et al. (2023). In particular, Lv et al. (2024); Liu

et al. (2025a) investigate the quantization challenges of SAM and introduce methods capable of producing per-channel quantized models with few data samples. Although PTQ avoids full retraining, the inability to adapt model parameters during quantization often leads to suboptimal results at ultra-low bit-widths. This limitation is highlighted in Figure 1, where state-of-the-art PTQ methods already exhibit significant performance degradation at 4-bit precision. The failure of PTQ at 4 bits motivates our work on a more robust, training-aware approach to achieve an accurate 2-bit model.

Quantizing SAM2 is nontrivial, particularly for the image encoder, a hierarchical transformer Ryali et al. (2023) which accounts for over 90% of the total parameters (in the B+ model) and poses a significant challenge for ultra-low-bit quantization (see Section 3). This difficulty stems from two main issues: the linear layer weight matrices exhibit values that deviate significantly from the distribution center (introducing outliers), and the activation output displays heavy-tailed distributions. Both of these characteristics severely degrade performance when uniform quantization schemes are applied.

To address these challenges and achieve 2-bit precision, we propose QAT-SAM, a complete QAT pipeline. As shown in Figure 2, our approach introduces two novel components to tackle the critical issues of high weight variance and activation outliers: an initialization method called **Variance-Reduced Calibration (VRC)**, and a QAT algorithm named **Learnable Statistical Clipping (LSC)**. This two-step approach first prepares the model for quantization and then robustly trains it, ensuring high accuracy even at 2-bit precision. Our main contributions are as follows:

- We introduce QAT-SAM, to the best of our knowledge the first quantization pipeline for the SAM2 architecture. Our method substantially improves QAT robustness at extreme bit-widths over the strongest prior QAT baselines, with gains up to 7.3 ppt in mIoU. We position the result against PTQ methods reported on the original SAM as a cross-generation reference (Figure 1).

- We propose a novel light calibration method, VRC, that reduces the variance of the encoder's weight distributions using a small batch of images. This procedure significantly lowers the initial quantization error, providing an improved starting point for the subsequent training.

- We develop a tailored QAT pipeline that leverages the calibrated encoder (from VRC) and a novel algorithm, LSC. The LSC grounds the quantization range using robust EMA-tracked statistical estimates $(\mu, \sigma)$. This hybrid design introduces only a single, learnable coefficient $(k)$ per quantizer, enabling stable ultra-low-bit convergence.

The paper is organized as follows: Section 2 describes the preliminaries and related work, Section 3 shows the QAT-SAM pipeline, Section 4 presents our experimental setup and results, and Section 6 concludes our findings.

## 2 Preliminaries and Related Work

### 2.1 Model Quantization

Quantization is the process of approximating a continuous or high-resolution discrete domain by a finite set of representative levels, to optimize computational efficiency and memory usage with controlled performance degradation. The quantization process can be formulated as:

$$\boldsymbol{X}_q = \text{clip}\left(\text{round}\left(\frac{\boldsymbol{X}}{s}\right) + z, \ 0, \ 2^b - 1\right). \tag{1}$$

Where **X** is a high-precision input tensor (e.g., FP32). The process maps **X** to the low-bit integer tensor $\mathbf{X}_q$ using the scaling factor $s$ and the integer zero-point $z$. Specifically, $s$ defines the step size of the quantization levels, while $z$ acts as an offset ensuring that the real-valued zero is mapped exactly to an integer. The bit-width $b$ determines the cardinality of the discrete set, typically $[0, 2^b - 1]$ for unsigned quantization. The clipping function is mathematically necessary to constrain the projected values round$(\mathbf{X}/s) + z$ to this finite integer domain. These parameters can be defined as scalars $(s, z \in \mathbb{R})$ for per-tensor quantization, or vectors

$(\boldsymbol{s}, \boldsymbol{z} \in \mathbb{R}^d)$ for per-channel quantization or a tensor ($\mathbf{S}, \mathbf{Z} \in \mathbb{R}^{d_1 \times d_2 \times \cdots}$) for per-group quantization. We focus on per-channel uniform quantization, which offers a good balance between efficiency and complexity.

## 2.2 Post-Training Quantization (PTQ)

Post-training quantization (PTQ) enables efficient deployment of large pretrained models by applying quantization without retraining the full network. Scale $s$ and zero-point $z$ are typically computed from calibration data. To improve accuracy, various PTQ methods introduce transformations to reduce quantization error: AdaRound Nagel et al. (2020) and BrecQ Li et al. (2021) optimize rounding and reconstruction, respectively. QDROP Wei et al. (2023) further refined this by adding a Dropout-inspired regularization, making weight optimization more robust to activation quantization errors. Other approaches leverage Hessian-based and calibration-free formulations like GPTQ Frantar et al. (2022) and HQQ Badri & Shaji (2023), or minimizing weight outliers via $L_\infty$-based regularization as in MagR Zhang et al. (2024). Activation outliers, common in transformers Dettmers et al. (2022); Wu et al. (2024), further complicate quantization. Techniques such as SmoothQuant Xiao et al. (2023), SpinQuant Liu et al. (2025b), and QuaRot Ashkboos et al. (2024) mitigate this by shifting or rotating activations to promote accurate PTQ.

In the vision domain, PTQ4ViT Yuan et al. (2022) tackles non-Gaussian activations in vision transformers. PTQ4DiT Wu et al. (2024) introduces channel-wise salience balancing to mitigate salient-channel and temporal activation outliers in Diffusion Transformers. PTQ4SAM Lv et al. (2024) is the first PTQ method tailored to SAM, utilizing bimodal integration and adaptive granularity quantization to transform challenging activation distributions.

## 2.3 Quantization Aware Training (QAT)

In Quantization Aware Training (QAT), the quantization process is simulated during training through Equation 1, allowing the network to adapt to quantization noise and minimize accuracy degradation. The scale $s$ and zero point $z$ can be treated as learnable parameters or are dynamically adapted, and the model weights themselves are updated through backpropagation to account for quantization effects. This joint adaptation allows the model to recover from quantization-induced distortions and is essential to preserve accuracy under low-bit regimes. Formally, QAT introduces non-differentiable quantization into the training loop, typically handled using straight-through estimators (STE) Bengio et al. (2013) to enable gradient-based optimization.

Several QAT methods have been proposed to improve accuracy in low-precision regimes. One of the first and most influential works is the MinMax-based QAT approach Jacob et al. (2018), which uses affine quantization and simulated integer operations during training, enabling efficient 8-bit deployment with integer-only inference. Another effective method is DoReFa-Net Zhou et al. (2016), which introduces quantization of weights, activations, and gradients using deterministic functions. However, its use in modern transformer architectures is limited due to the inability to quantize models with large activation ranges Li et al. (2022). PACT Choi et al. (2018) introduced a learnable clipping parameter for activations, enabling better control of dynamic range. Nevertheless, this introduces additional parameters and gradients, complicating the deployment.

LSQ Esser et al. (2019) is widely regarded as one of the most robust QAT methods, learning the quantization step size via backpropagation. Despite its introduction in 2019, LSQ remains a cornerstone of modern quantization research; it continues to serve as a primary baseline and foundational technique in recent high-impact works, such as the ParetoQ framework Liu et al. (2025c), which utilizes learnable scales to establish state-of-the-art scaling laws for extremely low-bit regimes. LSQ+ Bhalgat et al. (2020) finally improves training stability by introducing a log-based gradient formulation for the learnable scale, which prevents the scale from becoming zero and stabilizes the training process.

QAT remains underexplored in transformer-based vision architectures, where challenges such as weight oscillation, activation outliers, and attention-specific sensitivity can severely impact low-bit performance. Existing studies Liu et al. (2023); Huang et al. (2024) focus solely on LSQ, without evaluating alternative strategies or providing broader comparisons between quantization methods.

## 3 Method

### 3.1 Challenges in Quantizing SAM2

The SAM2 architecture employs a Hiera-based encoder Ryali et al. (2023) using hierarchical token merging for multiscale representation. This structure introduces distinct quantization challenges not found in its predecessor or the more recent SAM3 Carion et al. (2025). While the original SAM and the heavyweight 840M-parameter SAM3 maintain high redundancy to absorb precision noise, SAM2 optimizes efficiency through aggressive downsampling and reduced parameter counts. This architectural shift causes quantization errors to amplify into structural failures rather than being absorbed.

As a preliminary investigation, we analyze explicit value clipping on weight matrices to mitigate outliers by truncating extreme values to increase resolution in dense regions. While this results in an initial accuracy drop, the preservation of core predictive behavior indicates that these outliers are non-essential for effective functioning. We observe the same behavior for activations, which are even more difficult to quantize on low bits because of their higher value range. The complete findings and supporting analysis for this preliminary investigation, and the quantitative quantization comparison with ViT based SAM are detailed in Appendix C.

Our method is built on a key insight: since outliers can be clipped without destroying the model's predictive power, the problem becomes optimizing the remaining, dense part of the distribution. We therefore propose a two-stage pipeline: first, we apply **Variance-Reduced Calibration (VRC)** (Section 3.2), a novel initialization that narrows the encoder's weight distributions. This provides a low variance, so a lower quantization error starting point. Second, we apply **Learnable Statistical Clipping (LSC)** (Section 3.3), a robust QAT algorithm that learns optimal clipping thresholds for both calibrated weights and activations.

### 3.2 Variance-Reduced Calibration (VRC)

Although weight calibration is common in PTQ, the idea of introducing a dedicated preprocessing step before QAT has received little attention Li et al. (2022). In this work, we show that our calibration algorithm, named VRC can improve performance, particularly in ultra-low bit-width regimes.

While initially designed as a precursor to our clipping-based QAT, our calibration minimizes weight variance to compact the distribution, serving as a versatile preprocessing step universally applicable to any standard PTQ or QAT method. This allows for a smaller quantization step size, directly reducing the quantization error for the majority of the weights. We employ a computationally efficient, closed-form minimization of the Frobenius norm using a small calibration set. This approach is theoretically grounded: since the weight means are generally close to zero (with an average value of $-2 \times e^{-4}$ for linear layers in the B+ encoder, and a maximum of $-1 \times e^{-3}$), the Frobenius norm serves as an accurate proxy for variance, i.e.,

$$\text{Var}(\boldsymbol{W}) = \frac{1}{N} \sum_{i,j} (w_{ij} - \mu)^2 \xrightarrow{\mu=0} \frac{1}{N} \|\boldsymbol{W}\|_F^2.$$

However, any modification of a fully trained linear layer weight matrix risks degrading the accuracy of the model. A straightforward but crucial observation is that the accuracy degradation in the final model is directly related to the discrepancy between the output of the linear layers before and after quantization, rather than the quantization error of the matrix elements themselves. Any changes of $\boldsymbol{W}$ will therefore need to balance the compression gains due to quantization with the resulting errors at the output of linear layers.

Based on this insight, we propose a quantization procedure aimed at optimizing this trade-off through an initial calibration step that projects the trained weights onto the subspace spanned by a calibration batch of data samples, which ensures that the layer's outputs are preserved on these known inputs. In particular, we propose to perform this projection learning the *Moore-Penrose Pseudoinverse* between the input activations and the output of the linear layer while running a forward pass of the original trained model on the calibration samples.

While the residual between pre- and post-calibration output activations is guaranteed to be minimized only on the calibration samples, the Moore-Penrose Pseudoinverse has additional properties that make it

suitable as a preprocessing step before quantization. The fact that it allows us to control the singular values of the calibrated weight matrix (see later the discussion on Tikhonov regularization) implies that the calibration process "cleans" its spectrum by eliminating components that are irrelevant, unstable, or poorly supported by the calibration data, leading to a simpler and more robust solution. As a result, the modified layer maintains essential functionality on the calibration inputs while promoting smoother and potentially more general behavior elsewhere, which is particularly important for robustness under quantization or other perturbations.

The selection of calibration data is critical: larger batches improve faithfulness to the original model, while smaller ones reduce memory and runtime costs. Moreover, choosing representative frames is equally important because stacking ill-conditioned inputs in the calibration set can collapse the key directions learned during training, harming generalization. As a result, the layer may perform well on calibration samples, but fail on unseen or diverse data. Henceforth, calibration inputs must be diverse, well-conditioned, and representative of the operational domain of the model.

The condition number of activations reflects the calibration input quality: low values indicate stable and rich subspaces. However, aggregating inputs presents a risk: a single ill-conditioned sample can propagate instability to the entire batch, compromising the overall projection quality. Since pre-filtering inputs is computationally infeasible, we employ regularization to suppress ill-conditioning, enabling random input selection.

A linear layer is defined by $f(\boldsymbol{X}, \boldsymbol{W})$. It maps an input matrix $\boldsymbol{X} \in \mathbb{R}^{B \times d_{\mathrm{in}}}$, where $B$ is the batch size and $d_{\mathrm{in}}$ is the input feature dimension, to an output matrix $\boldsymbol{Y} \in \mathbb{R}^{B \times d_{\mathrm{out}}}$, where $d_{\mathrm{out}}$ is the output feature dimension, through an affine transformation defined as $f(\boldsymbol{X}, \boldsymbol{W}) = \boldsymbol{X}\boldsymbol{W}^{\top} + \boldsymbol{b}$, with $\boldsymbol{W} \in \mathbb{R}^{d_{\mathrm{out}} \times d_{\mathrm{in}}}$ and $\boldsymbol{b} \in \mathbb{R}^{d_{\mathrm{out}}}$. We consider $\boldsymbol{b} = \boldsymbol{0}$ for the following analysis, without loss of generality.

We compute the minimum Frobenius norm weight matrix $\hat{\boldsymbol{W}}$ for the batch $\boldsymbol{X}$ using the Moore-Penrose pseudoinverse Penrose (1955). For $\boldsymbol{X} \in \mathbb{R}^{B \times d}$ and $\boldsymbol{Y} \in \mathbb{R}^{B \times p}$, among all matrices $\boldsymbol{W} \in \mathbb{R}^{p \times d}$ satisfying $\boldsymbol{Y} = \boldsymbol{X}\boldsymbol{W}^{\top}$, the unique matrix minimizing the Frobenius norm $\|\boldsymbol{W}\|_F$ is

$$\hat{\boldsymbol{W}}^{\top} = \boldsymbol{X}^{\dagger}\boldsymbol{Y}, \tag{2}$$

where $\boldsymbol{X}^{\dagger}$ denotes the Moore–Penrose pseudoinverse of $\boldsymbol{X}$.

To mitigate the instability caused by poorly conditioned calibration samples, we applied the Tikhonov regularization during the projection. From Equation 2 the new formulation is derived:

$$\hat{\boldsymbol{W}}^{\top} = (\boldsymbol{X}^{\top}\boldsymbol{X} + \lambda \boldsymbol{I}_d)^{-1}\boldsymbol{X}^{\top}\boldsymbol{Y} \tag{3}$$

where $\lambda \in \mathbb{R}^{+}$ is the cutoff hyperparameter. We set $\lambda$ based on the singular values of the activation matrix to ensure stability during pseudoinverse computation. When the matrix is ill-conditioned but not rank-deficient, $\lambda$ is chosen as a small multiple of the smallest non-zero singular value. In cases where the smallest singular value indicates an effective rank deficiency (e.g., in low-precision formats like `bfloat16`), we instead select the smallest stable singular value and scale it accordingly. In practice, $\lambda = \lambda_0 \sigma_*$, where $\sigma_*$ is the smallest singular value that avoids rank deficiency, and $\lambda_0$ ranges between one and five. In Appendix A we present an analysis of our calibration method and the role of $\lambda$ in Equation 3.

The pseudoinverse formulation in Eq. 2 is regime-agnostic: when the calibration set is small ($B < d_{\mathrm{in}}$) the system is underdetermined and Eq. 2 selects the minimum-Frobenius-norm solution among the infinitely many zero-residual minimizers, while when $B > d_{\mathrm{in}}$ it reduces to the unique least-squares solution. With our default $n=50$-image calibration the relevant layers fall into the overdetermined regime ($B \gg d_{\mathrm{in}}$ from stacked-token batches), so the "minimum Frobenius norm" property is vacuous in this specific setting and variance reduction comes from the Tikhonov term in Eq. 3.

Figure 3 quantifies the efficacy of our VRC method on the SAM2.1-B+ model (using $\lambda_0 = 2.0$). The chart plots the average percent reduction in standard deviation across the linear layers within each transformer block. The dotted blocks contain layers with negative gains due to high instability in the input activations, in which the selected $\lambda_0$ is not sufficient to reduce the variance. In this case, the algorithm applies a fallback to the original layer, resulting in the normal colored block. Most blocks achieve a significant reduction of

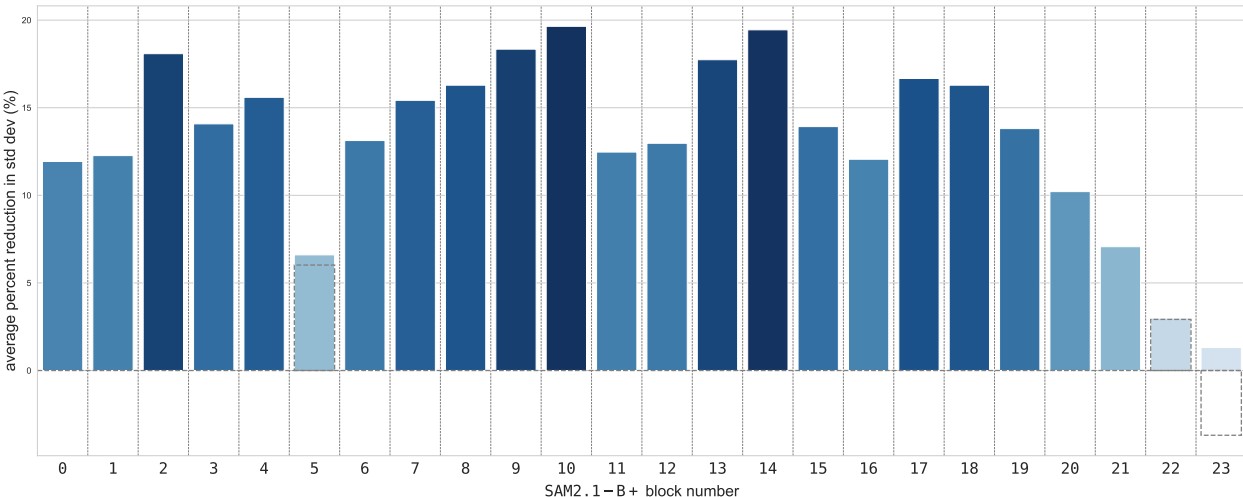

Figure 3: Impact of VRC on SAM2.1-B+ image encoder weight distributions for $\lambda_0 = 2.0$. Each bar shows the reduction in standard deviation averaged across all linear layers within one transformer block. Most blocks fall in the 10–20% range. The per-layer view (no averaging) is given in Figure 6, where individual layers reach up to 38.7%. This compresses the dynamic range, critically lowering the initial quantization error for the subsequent QAT.

more than 10%, with some approaching 20%. By systematically reducing the weight variance, the VRC compresses the dynamic range of the distributions. This reduction in the initial quantization error is critical for providing a stable and low-error starting point for the subsequent QAT phase. We report the calibrated encoder layer-by-layer in Appendix A.

### 3.3 Learnable Statistical Clipping (LSC)

Although learnable QAT methods like LSQ Esser et al. (2019) and LSQ+ Bhalgat et al. (2020) are effective, their "free-floating" learnable parameters (e.g., step-size $s$ and offset $z$) are not statistically grounded. This is a known source of instability, particularly in the per-channel, low-bit regime, where the large number of sensitive parameters leads to instability or variance, requiring a great deal of hyper-parameter tuning and initialization Bhalgat et al. (2020).

To solve this, we introduce **Learnable Statistical Clipping (LSC)**, a hybrid method that replaces these unstable learnable parameters with robust, statistically-grounded estimates. LSC anchors the quantization bounds to the data's true distribution, leaving only a single, well-behaved parameter to be optimized for each quantizer.

First, we track the first two moments of the distribution using an Exponential Moving Average (EMA), analogous to the robust estimators in Batch Normalization Ioffe & Szegedy (2015). For weights, we track, on each layer, the per-channel batch statistics $\hat{\mu}_{w,c}^{(t)}$ and $\hat{\sigma}_{w,c}^{(t)}$ at iteration $t$:

$$\mu_{w,c}^{(t)} = \beta\,\mu_{w,c}^{(t-1)} + (1-\beta)\,\hat{\mu}_{w,c}^{(t)},$$
$$\sigma_{w,c}^{(t)} = \beta\,\sigma_{w,c}^{(t-1)} + (1-\beta)\,\hat{\sigma}_{w,c}^{(t)},$$

where $\beta \in [0, 1)$ is the momentum term. We apply the same statistical tracking principle at the per-tensor level for activations.

The impact of this temporal smoothing is different for the two components. Activation batch statistics $(\hat{\mu}_a^{(t)}, \hat{\sigma}_a^{(t)})$ are inherently noisy, as they are highly dependent on the input data. The momentum $\beta$ is therefore essential for providing a stable, time-averaged estimate of the activation distribution. In contrast, weight statistics are much more static, especially as training converges. While the EMA still provides stabilization, its primary role for weights is to establish a robust estimate during the initial phase of QAT.

Second, we introduce a single, learnable, tensor-wide parameter $k \in \mathbb{R}^+$ for each quantizer. This learnable $k$ is optimized via backpropagation to find the optimal clipping range. This allows the network to learn the best balance between clipping error (the information lost from truncating outliers, which dominates when $k$ is small) and quantization error (the loss of precision from using wide buckets, which dominates when $k$ is large).

The final dynamic ranges of the quantized values for weights, as presented in Equation 1, are then defined symmetrically around the tracked mean:

$$\min_{w,c}^{(t)} = \mu_{w,c}^{(t)} - k_w \, \sigma_{w,c}^{(t)},$$
$$\max_{w,c}^{(t)} = \mu_{w,c}^{(t)} + k_w \, \sigma_{w,c}^{(t)}.$$

The per-tensor activation bounds $(\min_a^{(t)}, \max_a^{(t)})$ are computed analogously using their respective per-tensor statistics and learnable coefficient $k_a$.

This formulation is fundamentally more stable. A single layer in LSQ+ requires optimizing thousands of sensitive, independent per-channel parameters $(s_c, z_c)$ for weights alone. In contrast, our LSC replaces this entire set with robust, non-learnable statistics $(\mu_{w,c}^{(t)}, \sigma_{w,c}^{(t)})$ and just one learnable scalar, $k_w$. The optimization landscape is thus dramatically simplified. The gradients for $k_w$ and $k_a$ are well-behaved sums aggregated across all channels or tensor elements and are mediated by the stable $\sigma^{(t)}$ statistics.

# 4 Experiments

## 4.1 Experimental Setup

### 4.1.1 Task, Dataset, and Metrics

Our experimental setup resembles the one in the original SAM2 work Ravi et al. (2024). We apply the VRC and train the models on a subset of the SA-1B Kirillov et al. (2023) and SA-V Ravi et al. (2024) datasets. The SA-1B dataset is composed of 11M images and more than 1B masks, while SA-V contains 51k videos and 643K spatio-temporal segmentation masks. To reduce training time, we use a smaller subset of the dataset and still obtained high accuracy, demonstrating that we achieve performance without full retraining. From the original data, we subsample at random $\sim 8\%$ and $\sim 45\%$, respectively.

We evaluate our models on the semi-supervised Visual Object Segmentation (VOS) task, which involves segmenting a target object across a video given its ground-truth mask in the first frame. Following the SAM2 evaluation setup, we use SA-V val, SA-V test (both subsets of SA-V), and MOSE val from the MOSE dataset Ding et al. (2023), reporting $J\&F$ accuracy Pont-Tuset et al. (2017). For instance segmentation, we evaluate on the COCO2017 validation set Lin et al. (2014), using mean Intersection over Union (mIoU) across all 5,000 images. Moreover, following the same evaluation as PTQ4SAM Lv et al. (2024), we use the predicted boxes generated from DINO Zhang et al. (2022) as input prompt for our model, in order to calculate the mean Average Precision (mAP) on COCO2017.

### 4.1.2 Implementation

Our starting models are the checkpoints available from the SAM2 repository Meta AI Research (2024), named `sam2.1_hiera_tiny` (T), `sam2.1_hiera_small` (S), `sam2.1_hiera_base_plus` (B+). The implementation of the QAT pipelines is based on PyTorch 2.6.0 and CUDA 12.4, and all experiments are performed on NVIDIA A100 80GB GPUs (unless otherwise specified). The VRC is then performed using $n = 50$ input images, chosen uniformly at random from the SA-1B dataset. We forward the images through the original image encoder and record the activations. After stacking the tensors, we obtain the calibrated weight matrices by applying equation 3 with $\lambda_0 = 2.0$. More information on $\lambda_0$ and $n$ choice is given in Appendix A.1.

To assess our method, we compare QAT-SAM against the classical MinMax method Jacob et al. (2018), PACT Choi et al. (2018), and LSQ+ Bhalgat et al. (2020). We additionally include StatsQ Liu et al. (2023), a more recent, ViT-tailored baseline. Since standard LSQ+ (and LSQ Esser et al. (2019)) fails to converge

| Metric | Dataset | Methods | SAM2.1-B+ | | | SAM2.1-S | | | SAM2.1-T | | |
|---|---|---|---|---|---|---|---|---|---|---|---|
| | | | FP | W2A4 | W2A2 | FP | W2A4 | W2A2 | FP | W2A4 | W2A2 |
| *J&F* | SA-V Val Ravi et al. (2024) | MinMax Jacob et al. (2018) | 78.1 | 52.3 | 18.7 | 77.7 | 56.8 | 27.8 | 75.0 | 53.5 | 17.9 |
| | | PACT Choi et al. (2018) | | 54.7 | 36.4 | | 58.4 | 40.0 | | 56.3 | 43.4 |
| | | T_LSQ+ Bhalgat et al. (2020) | | 55.0 | 50.7 | | 58.1 | 51.7 | | 57.5 | 51.4 |
| | | StatsQ Liu et al. (2023) | | 60.7 | 46.7 | | 65.0 | 50.7 | | 63.8 | 48.4 |
| | | **QAT-SAM** | | **64.7** | **55.2** | | **65.1** | **56.0** | | **64.1** | **54.8** |
| | SA-V Test Ravi et al. (2024) | MinMax Jacob et al. (2018) | 78.2 | 56.8 | 21.2 | 76.6 | 59.9 | 29.7 | 75.0 | 57.1 | 19.5 |
| | | PACT Choi et al. (2018) | | 59.0 | 37.4 | | 60.6 | 43.1 | | 59.5 | 45.5 |
| | | T_LSQ+ Bhalgat et al. (2020) | | 59.0 | 53.3 | | 60.8 | 54.3 | | 60.0 | 55.5 |
| | | StatsQ Liu et al. (2023) | | 63.0 | 49.8 | | 66.5 | 52.8 | | 66.1 | 55.4 |
| | | **QAT-SAM** | | **64.3** | **58.3** | | **67.9** | **59.1** | | **67.1** | **59.3** |
| | MOSE Val Ding et al. (2023) | MinMax Jacob et al. (2018) | 73.7 | 55.3 | 26.5 | 73.5 | 59.1 | 33.5 | 70.9 | 58.8 | 26.2 |
| | | PACT Choi et al. (2018) | | 58.6 | 40.8 | | 60.6 | 43.7 | | 60.9 | 44.3 |
| | | T_LSQ+ Bhalgat et al. (2020) | | 56.0 | 52.9 | | 61.5 | 53.0 | | 60.2 | 54.7 |
| | | StatsQ Liu et al. (2023) | | 61.7 | 51.0 | | 66.5 | 54.3 | | 66.6 | 54.7 |
| | | **QAT-SAM** | | **64.1** | **57.1** | | **67.4** | **57.6** | | **66.8** | **58.2** |
| mAP | COCO2017 Lin et al. (2014) | MinMax Jacob et al. (2018) | 49.6 | 36.2 | 15.4 | 48.4 | 37.5 | 20.3 | 48.0 | 36.5 | 16.8 |
| | | PACT Choi et al. (2018) | | 38.3 | 25.2 | | 38.8 | 27.2 | | 38.7 | 28.2 |
| | | T_LSQ+ Bhalgat et al. (2020) | | 39.3 | 34.6 | | 40.2 | 31.4 | | 38.8 | 32.3 |
| | | StatsQ Liu et al. (2023) | | 44.2 | 35.3 | | 43.9 | 36.1 | | **43.8** | 33.0 |
| | | **QAT-SAM** | | **45.8** | **39.2** | | **44.3** | **36.1** | | 43.8 | **35.4** |
| mIoU | COCO2017 Lin et al. (2014) | MinMax Jacob et al. (2018) | 59.1 | 39.0 | 25.0 | 59.7 | 41.2 | 26.9 | 57.1 | 40.3 | 26.5 |
| | | PACT Choi et al. (2018) | | 41.2 | 32.0 | | 41.9 | 32.6 | | 38.7 | 28.2 |
| | | T_LSQ+ Bhalgat et al. (2020) | | 42.1 | 37.9 | | 42.6 | 36.4 | | 42.5 | 37.3 |
| | | StatsQ Liu et al. (2023) | | 42.2 | 31.2 | | 42.6 | 31.0 | | 42.1 | 32.6 |
| | | **QAT-SAM** | | **49.4** | **42.1** | | **47.3** | **39.6** | | **47.5** | **39.8** |

Table 1: Quantitative results on Video Object Segmentation (VOS) and Instance Segmentation. Our method is highlighted in grey. **Bold** indicates the best result for each configuration. The FP columns report the official SAM2 release numbers Ravi et al. (2024); the quantized columns are produced by our reduced training pipeline (see Appendix B.1.2).

with per-channel quantization Li et al. (2022), we implement a Tuned LSQ+ (T_LSQ+) baseline. This involves restricting the original LSQ+ to only activations quantization, tuning the learning rate for the step size $s$ and offset $z$, increasing observer calibration samplings, and employing per-channel MinMax for weights. To ensure a fair comparison, we evaluate LSQ+ against QAT-SAM in the W4A4 per-tensor schema. Even in this setting, QAT-SAM maintains a significant lead, outperforming LSQ+ by 9.7 ppt (VOS J&F) and 7.7 ppt (COCO mIoU). Full details, training pipelines, parameters, and losses are provided in Appendix B. Finally, we validate the stability of our training pipeline, observing a marginal standard deviation of 0.26 ppt in mIoU across independent runs.

## 4.2 Ultra-Low-Bit Quantization Performance

Table 1 summarizes the results of our experiments. The results show that our method consistently outperforms the other QAT algorithms by a large margin across almost each task, dataset, encoder size, and precision configuration. StatsQ improves over PACT and T_LSQ+ in the W2A4 configurations and becomes the strongest competitor there, while in the more aggressive W2A2 setting it falls behind T_LSQ+ on every model, leaving QAT-SAM as the only method that holds up under the extreme bit-width. For the B+ encoder, QAT-SAM demonstrates substantial improvement over the best baseline, achieving average gains on the video benchmark of 2.6 ppt (W2A4) and 4.6 ppt (W2A2). For instance segmentation, gains are 1.6 ppt (W2A4) and 3.9 ppt (W2A2) in mAP, and 7.2 ppt (W2A4) and 4.2 ppt (W2A2) in mIoU. The performance margin is reduced for the small and tiny encoders. This narrowing is expected because smaller models are inherently easier to quantize due to fewer outliers and narrower weight and activation distributions compared to the B+ configuration. This observation is attributed to the fact that the simpler distributions of the S and T encoders significantly boost the absolute performance of existing QAT baselines, thereby narrowing the gap with QAT-SAM. While T_LSQ+ substantially reduces the performance difference in the aggressive W2A2 configuration, QAT-SAM retains a significant lead. While the tiny configuration yields our smallest performance margin relative to baselines, representing the worst-case scenario for our gains, we still achieve

substantial improvements averaging 3.6 ppt on the video benchmark, 3.1 ppt in mAP, and 2.5 ppt in mIoU against T_LSQ+.

In a second experiment, we compare our QAT-SAM against post-training quantization (PTQ) methods—specifically PTQ4SAM Lv et al. (2024), BRECQ Li et al. (2021), and QDROP Wei et al. (2023) on the instance segmentation task. While these PTQ methods are not direct competitors to our approach, this comparison serves to illustrate the behavioral differences between architectures and to highlight the substantial performance impact enabled by QAT. As visualized in Figure 1, our lightweight QAT-SAM W2A4_B+ model achieves a 1.9 ppt mAP gain over the more than $8\times$ larger (in MB) PTQ4SAM-H W4A4 model. The complete comparison is reported in Appendix B.1.4.

In terms of GPU latency, since W2 kernels are not yet deployable, following literature we provided the level of BitOPs reduction obtaining 6.7x (compared to FP16) for W2A2 and 6.2x for W2A4 for the B+ encoder. Moreover, VRC latency and memory (RAM/VRAM) scale linearly with the number $n$ of calibration samples ($\approx$1.6GB/600MB per unit); selected $n = 50$ requires 263.7s, 81GB RAM, 29.9GB VRAM on Nvidia H100.

### 4.3 Ablation Studies

Table 2: Ablation study on VRC and LSC for B+ model. We report mAP for instance segmentation on COCO2017 dataset.

| Dataset | VRC ($\lambda_0 = 2.0$) | LSC | W2A2 |
|---|---|---|---|
| | $\times$ | $\times$ | 15.4 |
| COCO2017 Lin et al. (2014) | $\checkmark$ | $\times$ | 18.3 |
| | $\times$ | $\checkmark$ | 37.1 |
| | $\checkmark$ | $\checkmark$ | **39.2** |

Table 3: Ablation studies on the VRC for weight-only quantization for B+ encoder. The mAP for the image segmentation on COCO2017 and J&F for the VOS on SA-V val are reported.

| Dataset | Method | VRC ($\lambda_0$) | W3 | W4 |
|---|---|---|---|---|
| | | $\times$ | 24.9 | 47.5 |
| | HQQ Badri & Shaji (2023) | $\checkmark$(1.5) | 29.4 | **47.6** |
| | | $\checkmark$(2.0) | **29.6** | 46.0 |
| COCO2017 Lin et al. (2014) | | $\times$ | 15.7 | 46.3 |
| | MinMax Jacob et al. (2018) | $\checkmark$(1.5) | 19.6 | **46.9** |
| | | $\checkmark$(2.0) | **20.3** | 45.2 |
| | | $\times$ | 38.1 | **71.0** |
| | HQQ Badri & Shaji (2023) | $\checkmark$(1.5) | 41.2 | **71.0** |
| SA-V val Ravi et al. (2024) | | $\checkmark$(2.0) | **43.0** | 67.6 |
| | | $\times$ | 34.4 | 69.2 |
| | MinMax Jacob et al. (2018) | $\checkmark$(1.5) | 36.6 | **70.9** |
| | | $\checkmark$(2.0) | **39.5** | 64.4 |

Table 2 reports the contributions of VRC and LSC in trained networks. While the largest contribution stems from the LSC's management of activation outliers, VRC yields a 2.1 ppt gain. This improvement is statistically significant, as it substantially exceeds the observed training standard deviation of 0.26 ppt. The second ablation study evaluates the VRC calibration procedure in a post-training quantization setting using the B+ model with 3- and 4-bit weight-only quantization. We test VRC with two values of $\lambda_0$ on two methods: standard MinMax Jacob et al. (2018) and the Hessian-aware HQQ Badri & Shaji (2023). As detailed in Table 3, VRC consistently improves performance across both strategies. The gain is most significant in the 3-bit configuration, achieving a 4.6 ppt mAP gain for MinMax. In the 4-bit setting, the improvement is marginal ($\lambda_0 = 1.5$ offers a slight gain), confirming the expected trend: the contribution of VRC decreases as the bit-width becomes sufficient to represent weights accurately.

| Full Precision | T-LSQ+ Bhalgat et al. (2020) | PACT Choi et al. (2018) | **QAT-SAM (ours)** |
|---|---|---|---|

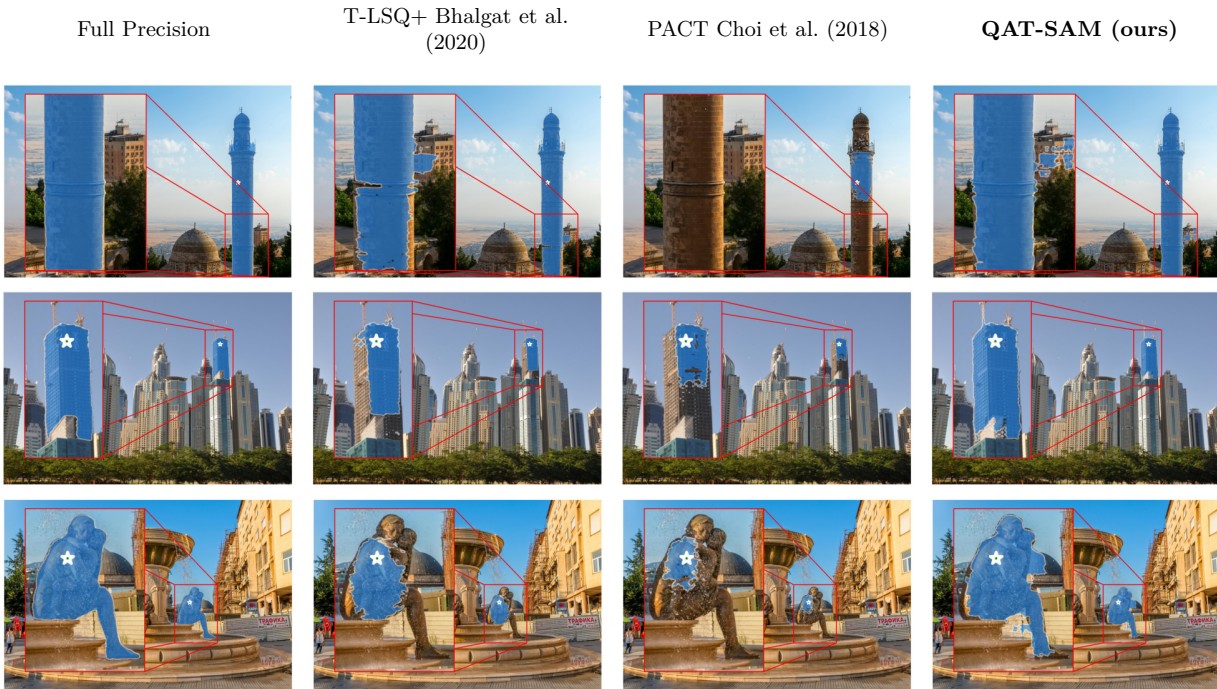

Figure 4: Qualitative results on W2A2 configuration for B+ encoder on promptable instance segmentation task.

### 4.4 Qualitative Results

Figure 4 presents a qualitative comparison of QAT-SAM versus full-precision SAM2.1 Ravi et al. (2024) and quantized baselines PACT Choi et al. (2018) and T-LSQ+ Bhalgat et al. (2020). Visually confirming our quantitative advantage, QAT-SAM demonstrates markedly improved segmentation quality. PACT consistently exhibits high levels of artifacts across all images and substantially fails to segment the input prompt. In contrast, T_LSQ+ fares marginally better, achieving only partial segmentation in some instances but invariably introducing noticeable artifacts. Our QAT-SAM yields segmentations closest to the full-precision model. Additional visual results and a video example are available in Appendix D.

## 5   Limitations

We outline three practical limitations of the present work. First, our evaluation concentrates on the Hiera encoder of SAM2; while VRC and LSC are not specific to that backbone, we have not validated them on the ViT-based encoders used by other SAM variants, whose weight and activation regimes differ. Second, the accuracy gains are most pronounced in the ultra-low-bit regime (W2A2 / W2A4) and remain substantial in the harder W4A4 per-tensor setting (Appendix B.1.3); for less constrained schemes, where standard QAT baselines already perform competitively, the relative advantage narrows. Third, on the deployment side, W2 inference kernels are not yet supported on commodity hardware, so the model-size reductions translate into storage and bandwidth savings today rather than wall-clock latency on typical accelerators; the BitOPs reductions in Section 4.2 are the hardware-agnostic proxy. The QAT phase itself incurs non-trivial compute (Appendix B.1) relative to the edge-deployment target, though this is a one-time, amortized cost.

## 6   Conclusions

We presented QAT-SAM, to the best of our knowledge the first quantized Segment Anything Model 2. Our method combined VRC, which reduces weight variance, and LSC, a robust QAT procedure that learns to clip

outliers. QAT-SAM significantly outperformed strong baselines on VOS and instance segmentation. Across all encoder sizes and quantization schemes evaluated, we observed average gains of 5.5 ppt J&F on the video benchmark, 4.7 ppt in mAP, and 4.5 ppt in mIoU, while qualitative results confirmed its effectiveness in preserving segmentation quality.

Future work could investigate the generalization of VRC and LSC to other architectures. A particularly promising direction involves exploring an adaptive VRC with a layer-specific $\lambda$ parameter, which enables a more fine-grained calibration to further optimize the accuracy-compression trade-off.

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

## Technical Appendices and Supplementary Material

This appendix provides more details to support the findings presented in the main paper. In Section A, we analyze the bias-variance trade-off introduced by the regularization hyperparameter $\lambda$ in the VRC, Section B outlines our training configuration and other results. In particular Section B.1.1 presents training loss curves, and Section B.1.2 discusses the differences between our training procedure and the original SAM2 model Ravi et al. (2024). In Section B.1.3 the LSQ+ Bhalgat et al. (2020) QAT algorithm is compared with our QAT-SAM in a per-tensor schema, and in Section B.1.4 we compare our performance with PTQ algorithm applied on SAM Kirillov et al. (2023). We present insights on the weight and activation distributions of SAM2 in Section C. Finally, Section D includes qualitative segmentation results to illustrate the effectiveness of our approach in both static images and a video.

## A  Variance-Reduced Calibration Details

To better analyze the behavior of our calibration method introduced in Section 3.2, we implement Equation 3 within the linear layer *cut.blocks.21.mlp.layers.1* of the image encoder in the B+ model. Calibration is performed by applying the regularized pseudoinverse formulation to batches of 50 images, chosen uniformly at random, while systematically varying the regularization hyperparameter $\lambda$. For each value of $\lambda$, we compute a calibrated weight matrix $\hat{\boldsymbol{W}}_\lambda \in \mathbb{R}^{d_{\text{out}} \times d_{\text{in}}}$.

To evaluate the effect of calibration, we compare the outputs of the original and calibrated layers using a separate evaluation batch of 10 images, denoted as $\boldsymbol{X}_{\text{eval}}$. The output discrepancy is quantified using the $L_2$ norm between the original projection $\boldsymbol{X}_{\text{eval}} \boldsymbol{W}^\top$ and the calibrated output $\boldsymbol{X}_{\text{eval}} \hat{\boldsymbol{W}}_\lambda^\top$. This metric provides a direct measure of the reconstruction error induced by the calibration process.

Figure 5a illustrates how the $L_2$ error varies with $\lambda$, revealing a clear bias-variance trade-off. For small regularization values ($\lambda < 10^{-3}$), the error is high and exhibits large variance across different calibration batches, indicating sensitivity to batch selection due to ill-conditioned activations. This suggests that when regularization is too weak, the inversion step amplifies noise, degrading output fidelity. On the opposite end, overly strong regularization ($\lambda > 1$) significantly alters $\hat{\boldsymbol{W}}_\lambda$, reducing its ability to generalize and match the behavior of the original layer. In the intermediate range ($10^{-3} \leq \lambda \leq 1$), the reconstruction error is minimized, indicating an optimal trade-off that stabilizes calibration without overfitting to the batch.

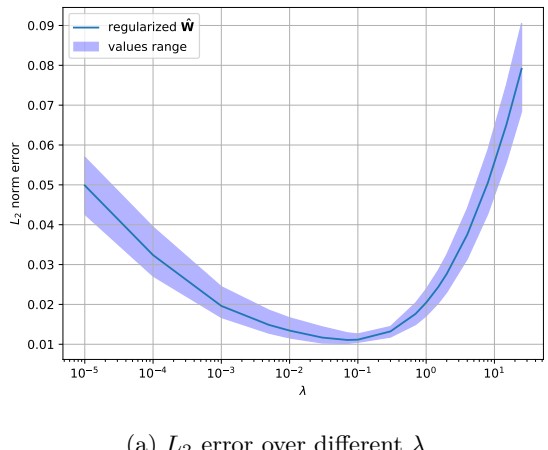
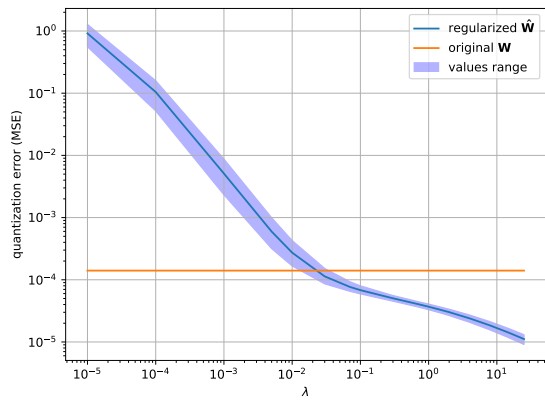

(a) $L_2$ error over different $\lambda$

(b) Quantization error (MSE) over different $\lambda$

Figure 5: Calibration error for the linear layer *cut.blocks.21.mlp.layers.1* from the image encoder of the B+ model. In subfigure (a), we compare the output of the original linear layer with the output produced using the calibrated weights $\hat{\boldsymbol{W}}_\lambda$, measured via the $L_2$ norm, across different values of the hyperparameter $\lambda$. In subfigure (b), we report the quantization error of $\hat{\boldsymbol{W}}_\lambda$ for varying $\lambda$, relative to the quantization error of the original weights $\boldsymbol{W}$.

To further assess the impact of calibration on quantization, we analyze the quantization error of $\hat{\boldsymbol{W}}_\lambda$, as shown in Figure 5b. The error is computed as

$$\frac{1}{d_{\text{out}} d_{\text{in}}} \| \hat{\boldsymbol{W}}_\lambda - \hat{\boldsymbol{W}}_{\lambda_Q} \|_F^2,$$

where $\hat{\boldsymbol{W}}_{\lambda_Q}$ is the quantized version of the calibrated weight matrix using uniform quantization. The orange line in the figure represents the quantization error when using the original weight matrix $\boldsymbol{W}$ instead of $\hat{\boldsymbol{W}}_\lambda$. We observe that counterintuitively for low $\lambda$ values, the quantization error is extremely high due to the poor conditioning of $\hat{\boldsymbol{W}}_\lambda$, which leads to numerical instability and large weight magnitudes. As $\lambda$ increases, particularly beyond $5 \times 10^{-2}$, the quantization error of the calibrated weights becomes consistently lower than that of the original weights. This suggests that regularization not only improves numerical stability, but also produces weight matrices that are better structured for quantization, exhibiting smaller dynamic ranges.

The selection of $\lambda$ is not straightforward. One option is to choose the value that minimizes the reconstruction error in Figure 5a, which provides a good match to the original layer outputs. However, this choice does not necessarily lead to the lowest quantization error, and its impact on initialization quality for quantized training may be limited. Conversely, setting $\lambda \approx 1$ improves the quantization behavior but can result in higher $L_2$ output error, compromising the layer's ability to generalize. This highlights the need to balance fidelity to the original outputs with improved robustness under quantization when selecting the regularization strength.

To select an appropriate value of $\lambda$ in practice, we adopt a rule based on the singular value spectrum of the input matrix. If the activation matrix is ill-conditioned (i.e., with a condition number greater than $10^2$) but not effectively rank deficient, we set $\lambda$ to five times the smallest non-zero singular value. In the case of low-precision formats such as `bfloat16`, matrices with condition numbers exceeding $10^3$ can already exhibit numerical instability. When the matrix is more severely ill-conditioned, with a condition number significantly greater than $10^3$, it is likely to be effectively rank deficient. In this case, we identify the smallest singular value $\sigma_*$ such that $\sigma_{\max}/\sigma_* \leq 10^3$, and set $\lambda = \lambda_0 \sigma_*$.

As a general rule, depending on the conditioning characteristics of the specific network and input data, $\lambda_0$ is chosen between two and five times the selected singular value. This strategy stabilizes the pseudoinverse computation while preserving the most meaningful directions in the activation space.

In Figure 6 we present the percentage reduction in standard deviation of all the layers after applying VRC to the B+ encoder when $\lambda_0 = 2.0$. In several blocks, some layers achieve a $\sigma$ reduction exceeding 30%, which significantly lowers the initial quantization error. As detailed in Section 3.2, the chosen $\lambda_0$ value may be insufficient for certain layers; we observe that three layers specifically fallback to their original uncalibrated state.

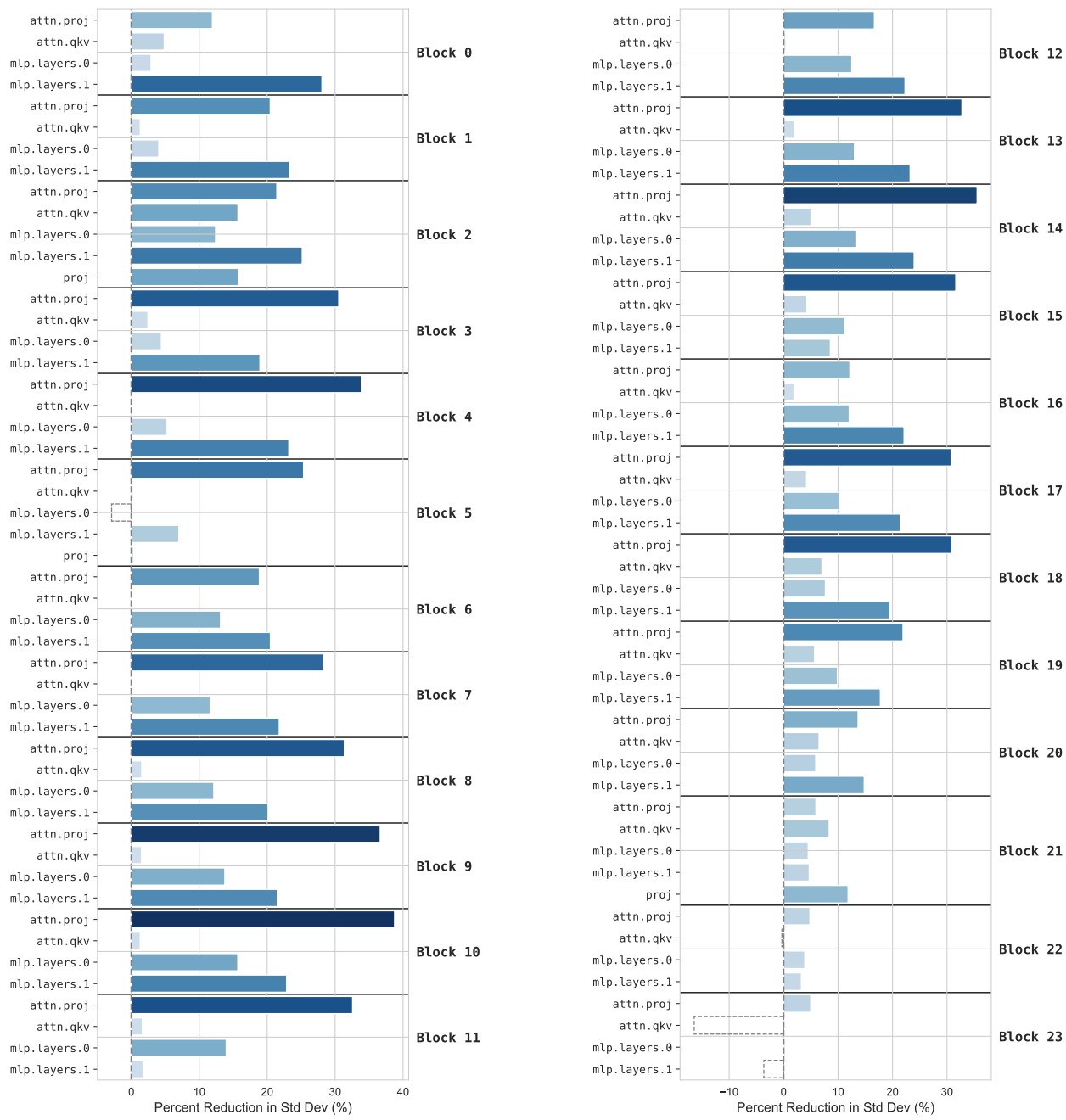

Figure 6: Impact of VRC on SAM2.1-B+ image encoder weight distributions for $\lambda_0 = 2.0$. Per-layer view, complementing the block-averaged Figure 3: VRC achieves a reduction up to 38.7% on individual layers, with a mean of 13.2% across all layers. The block-averaged 10–20% range in Figure 3 is consistent with this — within-block averaging smooths over the high-impact layers (toward 38.7%) and the low/fallback layers (toward 0% or below), yielding the narrower per-block band.

Table 4: Ablation of $\lambda_0$ and $n$. Values denote mIoU on COCO2017 and (quantization error reduction).

| n | $\lambda_0 = 1$ | $\lambda_0 = 2$ | $\lambda_0 = 3$ | $\lambda_0 = 5$ |
|---|---|---|---|---|
| 10 | 58.79 (5.6%) | 58.41(8.0%) | 58.00 (9.8%) | 57.19 (12.5%) |
| 30 | 59.03 (3.9%) | 58.92 (5.6%) | 58.13 (6.9%) | 58.61 (9.0%) |
| 50 | 59.13 (3.2%) | **59.06 (4.7%)** | 58.86 (5.8%) | 58.68 (7.6%) |
| 70 | 59.13 (3.0%) | 59.06 (4.4%) | 58.93 (5.5%) | 58.73 (7.2%) |

## A.1 VRC Ablations

In Table 4 we present the ablation over $\lambda_0$, and $n$. The results reveal a trade-off: higher $\lambda_0$ reduces quantization error (up to 12.5%) but degrades mIoU. Increasing calibration size to $n \geq 50$ stabilizes this accuracy loss. We selected $\lambda_0 = 2, n = 50$ as the optimal balance: it yields a 4.7% error reduction with a negligible drop in mIoU, ensuring the robust initialization critical for accuracy recovery in QAT.

## A.2 Adaptive VRC

We describe **Adaptive VRC (AVRC)**, a per-layer extension of the VRC procedure of Section 3.2. Standard VRC relies on two heuristics: a single global regularizer $\lambda_0$ shared across all linear layers, and a post-hoc fallback to the original weights whenever the calibrated weights increase the per-channel standard deviation. Figure 6 shows the cost of both: under a fixed $\lambda_0$ the per-layer variance reduction ranges from negative values (fallback fires) up to 38.7%, with mean 13.2%, so layers differ widely in the shrinkage they can absorb. AVRC replaces both heuristics with a single layer-local feasibility test driven by a calibration-grounded budget $\delta \in (0, 1)$.

Recall from Section 3.2 that for a layer with weights $\boldsymbol{W} \in \mathbb{R}^{d_{\text{out}} \times d_{\text{in}}}$, calibration inputs $\boldsymbol{X} \in \mathbb{R}^{B \times d_{\text{in}}}$, and outputs $\boldsymbol{Y} = \boldsymbol{X} \boldsymbol{W}^\top$, the regularized calibrated weights are

$$\hat{\boldsymbol{W}}_\lambda^\top = (\boldsymbol{X}^\top \boldsymbol{X} + \lambda \boldsymbol{I}_{d_{\text{in}}})^{-1} \boldsymbol{X}^\top \boldsymbol{Y}. \tag{4}$$

For a given layer, $\lambda$ is a 1D trade-off: small $\lambda$ preserves outputs but barely shrinks weights, large $\lambda$ shrinks aggressively but distorts outputs. We summarize each side by one scalar:

$$R(\lambda) = \frac{\|\boldsymbol{X} \hat{\boldsymbol{W}}_\lambda^\top - \boldsymbol{Y}\|_F^2}{\|\boldsymbol{Y}\|_F^2}, \qquad V(\lambda) = \|\hat{\boldsymbol{W}}_\lambda\|_F^2.$$

$R(\lambda)$ is the relative output deviation on the calibration set, normalized so it is dimensionless and comparable across layers; $V(\lambda)$ is a faithful proxy for the per-channel weight variance, since SAM2's weight means are negligibly small relative to their per-channel standard deviations. $R$ is non-decreasing and $V$ is non-increasing in $\lambda$, with $R(0) = 0$ in the overdetermined regime ($B \gg d_{\text{in}}$).

For each linear layer $\ell$ in the image encoder, AVRC proceeds as:

1. Cache the SVD $\boldsymbol{X}_\ell = \boldsymbol{U}_\ell \boldsymbol{\Sigma}_\ell \boldsymbol{V}_\ell^\top$.

2. Set $\lambda_\ell$ to the largest $\lambda \geq 0$ such that

$$R(\lambda) \leq \delta \quad \text{and} \quad V(\lambda) < \|\boldsymbol{W}_\ell\|_F^2. \tag{5}$$

   The first condition caps the relative output perturbation at $\delta$; the second guarantees the regularization is doing useful work.

3. If no $\lambda \geq 0$ satisfies (5), set $\hat{\boldsymbol{W}}_\ell = \boldsymbol{W}_\ell$. The fallback is now the explicit infeasibility of the criterion, not a separate heuristic.

Through the cached SVD both scalars admit a closed form,

$$R(\lambda) = \frac{\sum_i \left(\lambda/(\sigma_i^2 + \lambda)\right)^2 \|(\boldsymbol{U}_\ell^\top \boldsymbol{Y}_\ell)_i\|^2}{\|\boldsymbol{Y}_\ell\|_F^2}, \qquad V(\lambda) = \sum_i \frac{\sigma_i^2}{(\sigma_i^2 + \lambda)^2} \|(\boldsymbol{U}_\ell^\top \boldsymbol{Y}_\ell)_i\|^2,$$

so step 2 is a 1D root-finding problem solved by bisection on $\log \lambda$ over the layer-specific range bounded by the singular values of $\boldsymbol{X}_\ell$. Each evaluation is $O(d_{\text{in}})$ and requires no fresh matrix inverse, making AVRC essentially free on top of VRC.

$\lambda_\ell$ thus adapts to each layer's spectrum $\boldsymbol{\Sigma}_\ell$ and the coupling $\boldsymbol{U}_\ell^\top \boldsymbol{Y}_\ell$: layers whose weight mass lies in low-activation directions tolerate larger $\lambda_\ell$, while layers whose mass aligns with dominant activation directions select small $\lambda_\ell$ or the fallback.

Finally, AVRC is essentially insensitive to the specific calibration sample. Repeating it with $\delta = 0.1$ across 10 independently drawn calibration sets ($n{=}50$ images each) for the B+ encoder, the global per-layer standard-deviation reduction is $17.94\% \pm 0.16\%$ (range $17.67$–$18.24\%$) and the downstream COCO mAP is $48.0 \pm 0.1$ ppt. The dispersion is well below absolute performance differences between methods, so AVRC's per-layer choice of $\lambda_\ell$ is governed by the layer's spectrum rather than by the particular calibration images.

# B  Training and Other Results

## B.1  Training Parameters

| Configuration | Value |
|---|---|
| data | **SA-1B(8%) SA-V(45%)** |
| resolution | 1024 |
| precision | bfloat16 |
| epochs | **1** |
| optimizer | AdamW |
| optimizer momentum | $\beta_1, \beta_2 = 0.9, 0.999$ |
| gradient | clipping type: $L_2$, max: 0.1 |
| weight decay | 0.1 |
| learning rate (lr) | **img. enc.: 1e − 5**, other: $3.0e − 4$ |
| lr schedule | cosine |
| warmup | **no warmup** |
| layer-wise decay | 0.8 (T, S), 0.9 (B+) |
| image augmentation | hflip, resize to 1024 (square) |
| video augmentation | hflip, affine (deg: 25, shear: 20), colorjitter (b: 0.1, c: 0.03, s: 0.03,h: null), grayscale (0.05), per frame colorjitter (b: 0.1, c: 0.05, s:0.05, h: null), mosaic-2×2 (0.1) |
| batch size | 256 |
| drop path | 0.1 (T, S), 0.2 (B+) |
| mask losses (weight) | focal (20), dice (1) |
| IoU loss (weight) | $L_1$ (1) |
| occlusion loss (weight) | cross-entropy (1) |
| max. masks per frame | **image: 60**, video: 3 |
| # correction points | 7 |
| global attn. blocks | 5-7-9 (T), 7-10-13 (S), 12-16-20 (B+) |

Table 5: Hyperparameters and details of our QAT-SAM and baselines QAT training for the three image encoder sizes B+,S,T.

To ensure fairness, we used a consistent training and evaluation setup for both our QAT-SAM and the baseline approaches Minmax Jacob et al. (2018), PACT Choi et al. (2018), and T_LSQ+ Bhalgat et al.

(2020). The training procedure follows the one proposed in the SAM2 Ravi et al. (2024) paper, where the authors referred to the "full training" step. The details are reported in Table 5 where we highlight the configurations and hyperparameters that are different from the original work. All trained models use the same image and video data, with a fixed random seed to ensure reproducibility.

Compared to the original SAM2 training setup, we introduce several modifications to accommodate hardware constraints and reduce training time. First, we reduce the maximum number of masks per frame from 64 to 60 to fit within GPU memory limits on an NVIDIA A100 80GB. We also skip the warm-up phase, reduce the amount of input data, and limit training to a single epoch.

All training experiments are conducted using 8 NVIDIA A100 80GB GPUs, with a total training time of less than 35 hours for the $B+$ architecture.

We adjust the main learning rate from $4 \times 10^{-5}$ to $1 \times 10^{-5}$ and decrease the effect of the gradient scaler by halving its initial scale value. For the T_LSQ+ training, we reduce the scale and zero-point learning rate to $5 \times 10^{-8}$ to stabilize parameter convergence.

The initial value for the LSC parameter $k$ (Section 3.3) is set to $k = 2.5$ for B+ weights and activations, as well as for S and T activations. We use $k = 2.8$ for the S and T weights. Finally, for observer calibration before QAT, we infer 300 images for QAT-SAM, PACT, and MinMax, but increase this to 900 images for T_LSQ+ to capture its activation distributions more effectively.

### B.1.1 Losses Plots

We adopt the standard loss of SAM2 Ravi et al. (2024) as $\mathcal{L}_{\mathrm{SAM2}}$, which is composed of a linear combination of focal and dice losses for mask prediction, mean absolute error (MAE) loss for IoU prediction, and cross-entropy loss for object prediction. The weights are set in a fixed ratio of $20 : 1 : 1 : 1$, respectively.

Figures 7 and 8 present the loss curves for the W2A4 and W2A2 quantization settings. Specifically, Figure 7 shows the results for the B+ model and one configuration of the S model (W2A2), while Figure 8 includes the second S configuration (W2A4) together with both configurations for the T model.

In all settings and across all loss components, QAT-SAM demonstrates lower training losses compared to the MinMax, PACT, and T_LSQ+ baselines. In addition, our model converges faster as the difference in loss values, particularly for the dice and IoU components, increases progressively with each training step.

### B.1.2 Comparison with SAM2 training pipeline

The SAM2 is trained in two main stages followed by a fine-tuning procedure. The first stage, pre-training, involves training on static images using the SA-1B dataset. The second stage, called full training, uses a combination of images from SA-1B, videos from SA-V, and additional internal data that is not publicly available. Finally, to improve segmentation performance on long video sequences, the authors fine-tune the model by increasing the number of input video sequences from 8 to 16. To accommodate this within GPU memory constraints, the image encoder is frozen during this step. In our work, we reproduce the second stage, called "full training", focusing on a full architecture training with static image and video data while reducing both the data volume and the number of training steps with respect to the original paper. As demonstrated in Section B.1.1, our method learns faster than baselines and achieves accurate results with fewer data, thus reducing training time. However, a direct comparison with the original SAM2 model remains difficult due to differences in training procedures. We plan to explore a fully aligned comparison with the complete SAM2 pipeline in future work.

### B.1.3 Comparison with LSQ

We compare QAT-SAM with LSQ+ Bhalgat et al. (2020), a widely adopted and robust QAT algorithm known to learn quantization step sizes and zero point via gradient-based optimization. Although LSQ+ has shown strong results on various tasks, Table 6 shows that QAT-SAM significantly outperforms on all video and image benchmarks in the W4A4 (per-tensor) setting, achieving gains that exceed 10 ppt J&F for VOS, and 7.7 ppt mIoU for image segmentation. For the per-channel configurations, LSQ+, and LSQ Esser et al.

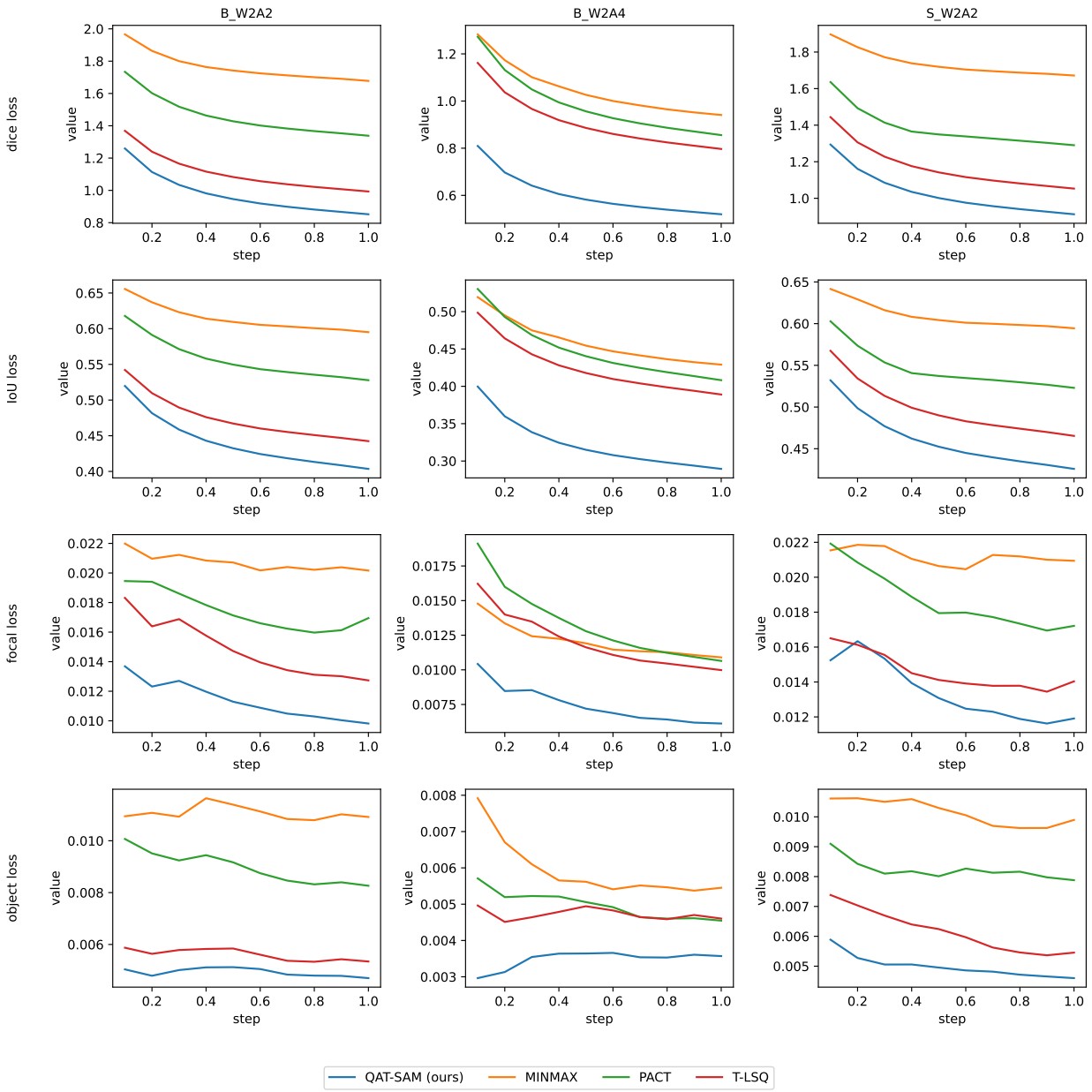

Figure 7: Training loss curves for image encoder size B+, S under two configurations W2A4 and W2A2. Each sub-plot contains a comparison between QAT-SAM and the baselines for the four losses: dice, IoU, focal, and occlusion. Our solution outperforms the baselines in all losses for all configurations.

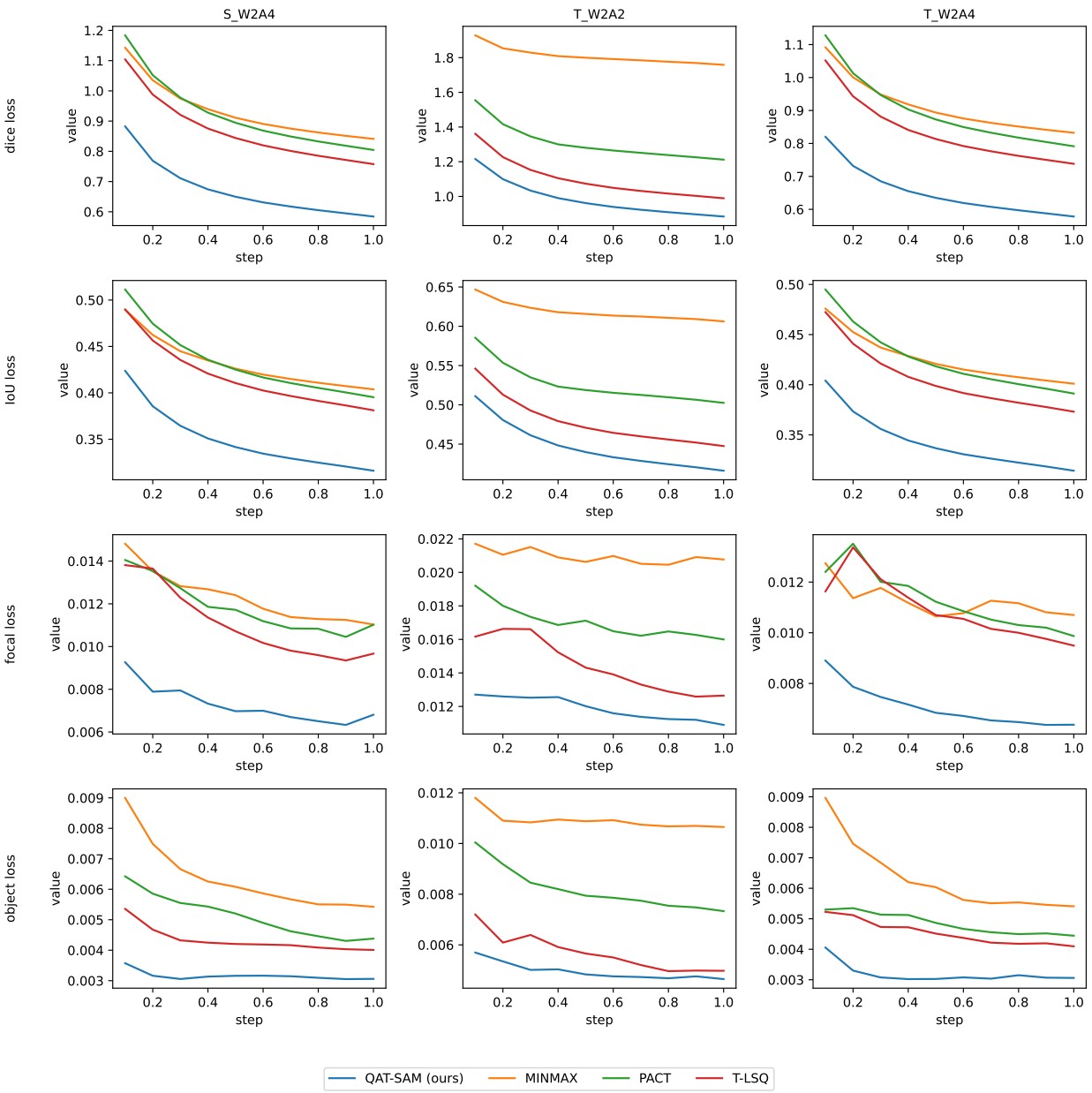

Figure 8: Continuation of the training loss curves from Figure 7 for image encoder size S, T under two configurations W2A4 and W2A2. Each sub-plot contains a comparison between QAT-SAM and the baselines for the four losses: dice, IoU, focal, and occlusion. Our solution outperforms the baselines in all losses for all configurations.

Table 6: Evaluation of the semi-supervised VOS and instance segmentation tasks for the B+ model and W4A4 (per-tensor) configuration. Our solution is compared with the LSQ+ Bhalgat et al. (2020) QAT algorithm.

| Metric | Dataset | Method | FP | W4TA4 |
|---|---|---|---|---|
| J&F | SA-V Val Ravi et al. (2024) | LSQ+ Bhalgat et al. (2020) | 78.1 | 56.4 |
| | | **QAT-SAM** | | **65.2** |
| | SA-V Test Ravi et al. (2024) | LSQ+ Bhalgat et al. (2020) | 78.2 | 56.6 |
| | | **QAT-SAM** | | **66** |
| | MOSE Val Ding et al. (2023) | LSQ+ Bhalgat et al. (2020) | 73.7 | 56.6 |
| | | **QAT-SAM** | | **67.5** |
| mAP | COCO2017 Lin et al. (2014) | LSQ+ Bhalgat et al. (2020) | 49.6 | 41.1 |
| | | **QAT-SAM** | | **47.9** |
| mIoU | COCO2017 Lin et al. (2014) | LSQ+ Bhalgat et al. (2020) | 59.1 | 44.9 |
| | | **QAT-SAM** | | **52.6** |

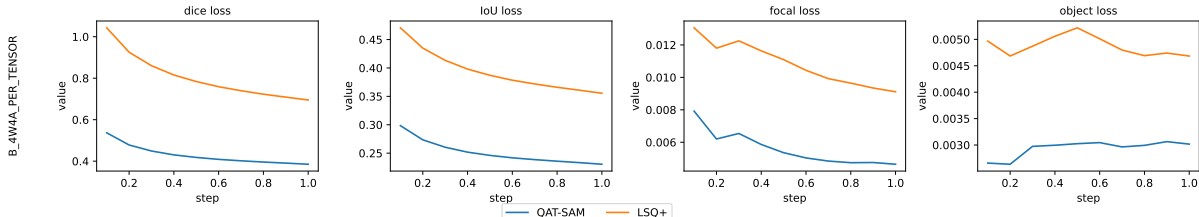

Figure 9: Training loss curves for image encoder size B+ under W4A4 (per-tensor) configurations. Each sub-plot contains a comparison between QAT-SAM and the LSQ+ Bhalgat et al. (2020) method for the four losses: dice, IoU, focal, and occlusion.

(2019) fail to converge, consistent with the limitations previously reported in Li et al. (2022); results are thus omitted. The training curves for LSQ+ are represented in Figure 9 and show that LSQ+ begins with a large performance gap compared to QAT-SAM; while the gap narrows slightly over time, LSQ+ still converges to significantly lower accuracy. This suggests that LSQ+ may require more training data or epochs to achieve competitive results, reinforcing the efficiency and stability of the QAT-SAM approach.

### B.1.4 Comparison with PTQ methods

We benchmark our QAT solution, QAT-SAM, against relevant PTQ quantization baselines. Our comparison includes PTQ4SAM Lv et al. (2024), an algorithm specifically developed for the original SAM architecture Kirillov et al. (2023). We also report two leading general PTQ schemes, QDROP Wei et al. (2023) and BRECQ Li et al. (2021), known for their strong general performance. Table 7 summarizes the superior accuracy-compression trade-off achieved by QAT-SAM. We show that our method improves robustness at ultra-low bit-widths over PTQ baselines reported on SAM, while delivering large size reductions. QAT-SAM achieves exceptional compression, with the QAT-SAM-B+ model requiring only 38.8 MB for storing the weights. This results in a $\sim 8.1$x reduction in model size compared to the largest baseline, PTQ4SAM-H (W4A4, 317 MB). Our smallest QAT-SAM-S and QAT-SAM-T configurations further reduce size to approximately 30 MB. Crucially, the QAT-SAM models retain accuracy far beyond similarly compressed or even larger baselines. The QAT-SAM-B+ (W2A4, 45.8 mAP) is competitive with the largest PTQ4SAM-H (W4A4, 43.9 mAP). Furthermore, QAT-SAM-B+ remains robust even in the most aggressive W2A2 setting, retaining 39.2 mAP, which surpasses the W4A4 performance of the larger PTQ4SAM-L (W4A4, 36.6 mAP).

Table 7: Comparison between QAT-SAM and PTQ methods based on the original SAM Kirillov et al. (2023). PTQ rows are evaluated on the original SAM (ViT-B/L/H); QAT-SAM rows on SAM2 (Hiera). The two halves differ in both quantization paradigm and backbone, so the comparison is illustrative of size-vs-accuracy trends across SAM generations rather than a same-backbone evaluation (cf. Figure 1).

| Encoder | Precision | Size W (MB) | Method | mAP | FP |
|---------|-----------|-------------|--------|-----|-----|
| ViT-H | W6A6 | 477 | PTQ4SAM Lv et al. (2024) | 48.7 | 49.1 |
| | | | QDROP Wei et al. (2023) | 48.3 | |
| | | | BRECQ Li et al. (2021) | 46.0 | |
| | W4A4 | 317 | PTQ4SAM Lv et al. (2024) | 43.9 | |
| | | | QDROP Wei et al. (2023) | 41.7 | |
| | | | BRECQ Li et al. (2021) | 17.6 | |
| ViT-L | W6A6 | 230 | PTQ4SAM Lv et al. (2024) | 48.3 | 48.6 |
| | | | QDROP Wei et al. (2023) | 47.5 | |
| | | | BRECQ Li et al. (2021) | 46.6 | |
| | W4A4 | 154 | PTQ4SAM Lv et al. (2024) | 36.6 | |
| | | | QDROP Wei et al. (2023) | 27.5 | |
| | | | BRECQ Li et al. (2021) | 12.3 | |
| ViT-B | W6A6 | 71 | PTQ4SAM Lv et al. (2024) | 40.4 | 44.5 |
| | | | QDROP Wei et al. (2023) | 38.9 | |
| | | | BRECQ Li et al. (2021) | 31.8 | |
| | W4A4 | 47 | PTQ4SAM Lv et al. (2024) | 14.4 | |
| | | | QDROP Wei et al. (2023) | 11.2 | |
| | | | BRECQ Li et al. (2021) | 3.6 | |
| Hiera-B+ | W2A4 | 38.8 | QAT-SAM | 45.8 | 49.6 |
| | W2A2 | 38.8 | | 39.2 | |
| Hiera-S | W2A4 | 32.0 | | 44.3 | 48.4 |
| | W2A2 | 32.0 | | 36.1 | |
| Hiera-T | W2A4 | 30.2 | | 43.8 | 48.0 |
| | W2A2 | 30.2 | | 35.4 | |

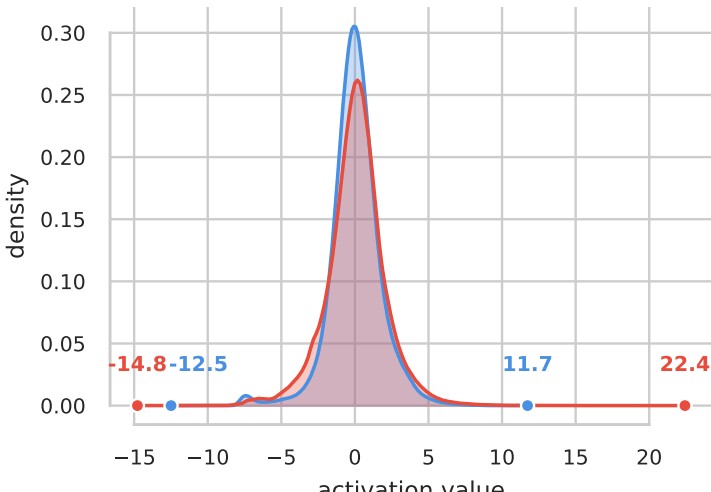

Figure 10: Comparison of the kernel density estimates across 50 random input generating the lowest (blue) and highest (red) total magnitude response in the *image_encoder.trunk.blocks.1.mlp.layers.0* layer of the B+ model. Markers indicate the precise minimum and maximum values, highlighting the shift in dynamic range between distinct inputs.

Table 8: COCO2017 Lin et al. (2014) mIoU using SmoothQuant Xiao et al. (2023) ($\alpha = 0.5$).

| Scheme | SAM-H | SAM-L | SAM2 B+ | VRC SAM2 B+ |
|--------|-------|-------|---------|-------------|
| FP | 58.41 | 58.23 | 59.10 | 59.10 |
| W6A6 | 57.70 (-0.71) | 57.66 (-0.57) | **51.00 (-8.10)** | 51.49 (-7.61) |
| W4A4 | 29.56 (-28.85) | 29.19 (-29.04) | **21.48 (-37.62)** | 21.88 (-37.22) |
| W3A8 | 53.22 (-5.19) | 48.26 (-9.97) | **10.58 (-48.52)** | 12.28 (-46.82) |
| W4A8 | 57.91 (-0.50) | 58.23 (-0.00) | **54.89 (-4.21)** | 57.38 (-1.72) |

## C   SAM2 Weight and Activation Distributions

As discussed in Section 3.1, the core challenge in quantizing SAM2 stems from extreme weight and activation distributions. The architectural shift from SAM's ViT to SAM2's Hiera encoder is, by itself, sufficient to break ViT-tuned quantization recipes: applying SmoothQuant under matched recipes, SAM1 loses 0.7 ppt at W6A6 versus 8.1 ppt for SAM2, and 5.2 ppt vs. 48.5 ppt at W3A8 (Table 8).

We quantify the severity of the underlying weight outliers in the B+ encoder by analyzing the ratio of the standard deviation ($\sigma$) to the absolute maximum value ($\max(|\boldsymbol{W}|)$). On average, the absolute maximum value is located at $13.6\sigma$ with a peak of $39.0\sigma$. These results confirm that high-magnitude outliers reside far from the central mass of the distribution, consuming a disproportionate share of the quantization range and necessitating specialized treatment.

Furthermore, we demonstrate that the quantization challenge extends significantly to activations. Beyond exhibiting large value ranges, we observe substantial distribution shifts within the same layer during inference. This inherent statistical instability, which we visualize in Figure 10, underscores the critical need for a stabilization mechanism like momentum-stabilized clipping. Building upon these observations, we investigate fixed clipping for weights and activations as an initial outlier mitigation strategy. This involves truncating values based on statistical bounds, specifically using the rule

$$\mu \pm \alpha\sigma, \tag{6}$$

Table 9: Ablations of value $\alpha$ from Equation 6 using model B+ and W2A4 configuration.

| Dataset | Clipping ($\alpha$) | VRC | mIoU |
|---|---|---|---|
| | $\times$ | $\times$ | 39.0 |
| COCO2017 Lin et al. (2014) | 2.6 | $\times$ | 43.3 |
| | 2.6 | $\checkmark$ | 44.7 |

where $\mu$ and $\sigma$ are the distribution mean and standard deviation, and $\alpha$ is a fixed scaling factor. We apply Equation 6 on a MinMax approach where we calculate the statistics per-tensor for activations and per-channel for weights, then applying a fixed $\alpha = 2.6$. Table 9 shows the results of this experiment, and confirms the effectiveness of statistical clipping, with a 4.3 ppt improvement in accuracy over the unclipped baseline. Furthermore, our VRC enhances the performance of the MinMax approach, demonstrating additional 1.4ppt gains beyond those achieved by simple static bounds.

## D   Qualitative Results

We have extensively benchmarked the original variants of SAM2, the MinMax baseline, and our QAT-SAM algorithm. The variation includes different architectures, prompts, and quantization configurations for a fair comparison in real-world scenarios. Figure 11, Figure 12, and Figure 13 show the impact of adding more input points as prompt. We observe that our algorithm provides cleaner masks compared to PACT, and T_LSQ+ algorithms with the same quantization scheme. We also note that by providing one prompt point only, typically all variants of the algorithms tend to provide a clean mask. However, and that is a well-studied behavior of the original SAM model, this mask might be only a part of the user's intended selection, such as the front of the racing car as in Figure 12, or a single garlic bulb as in Figure 13. Henceforth, interesting examples include those that require multiple prompts to select the entire object. Those 3-point or 5-point prompts trigger differences among the quantized models. Compared against PACT, and T_LSQ+, our 2-bit models require fewer prompt points to recover a larger fraction of the entire object, producing a cleaner and better result.

Figure 14, Figure 15 show the entire family of models including SAM2 base plus, small, and tiny architectures at full precision, as well as all variants of PACT, T_LSQ+, and our QAT-SAM models with 2-bit quantization for weights and 2 and 4 bits for activations. We first analyze the visual results prompted by a single point (Image 14). In this challenging scenario, many baseline models erroneously segment only the object's head. Our QAT-SAM family (W2A4 variants) and the B+ W2A2 model yield segmentation maps comparable to the full-precision network, whereas other baselines struggle significantly. We examine the second image (prompted by three points), where nearly all models correctly identify the target object. However, a closer inspection reveals distinct qualitative differences between our proposed QAT-SAM solution and the competing baselines.

Finally, Figures 16 and 17 present a qualitative comparison on a video from the SA-V test set for the semi-supervised Visual Object Segmentation (VOS) task. The models included in this comparison are SAM2.1, the baselines, and our proposed QAT-SAM for the W2A2 schema. The first column of Figure 16 shows the initial ground-truth mask used to prompt the models, as required in the semi-supervised VOS setup. Although T_LSQ+ generally succeeds in segmenting the target object, QAT-SAM produces cleaner masks with fewer border artifacts compared to the baselines.

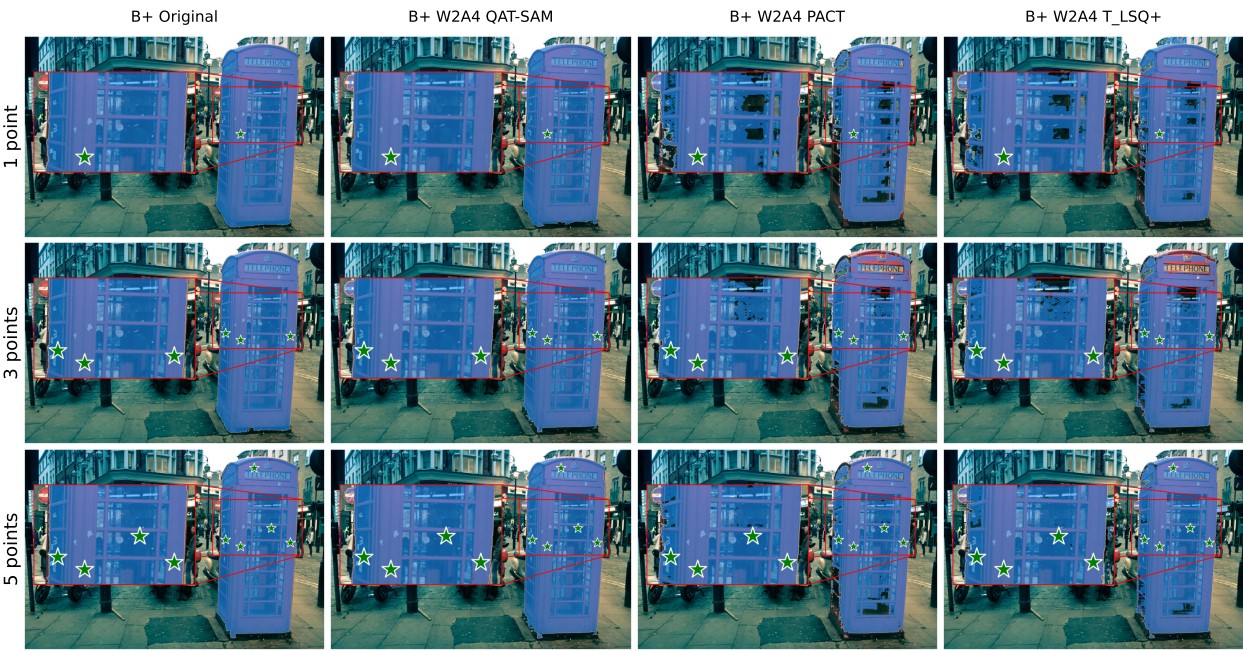

Figure 11: Qualitative results of SAM2, QAT-SAM(ours), PACT, and T_LSQ+ on the base plus architecture. From top to bottom, we add 1, 3, and 5 prompt points.

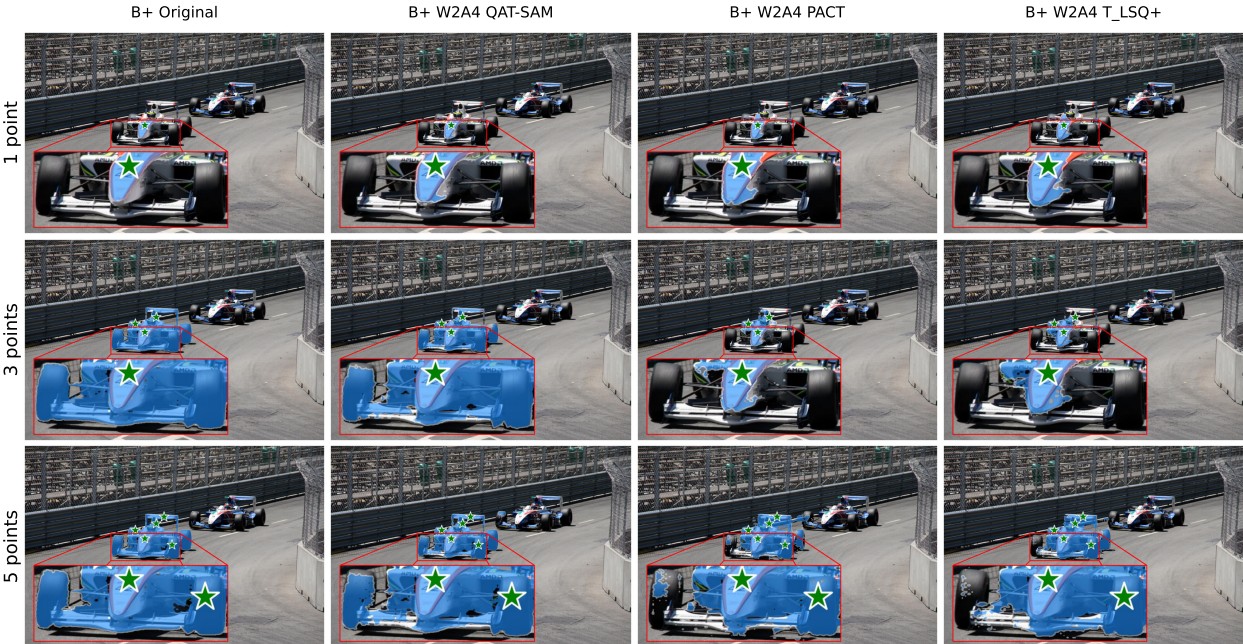

Figure 12: Qualitative results of SAM2, QAT-SAM(ours), PACT, and T_LSQ+ on the base plus architecture. From top to bottom, we add 1, 3, and 5 prompt points.

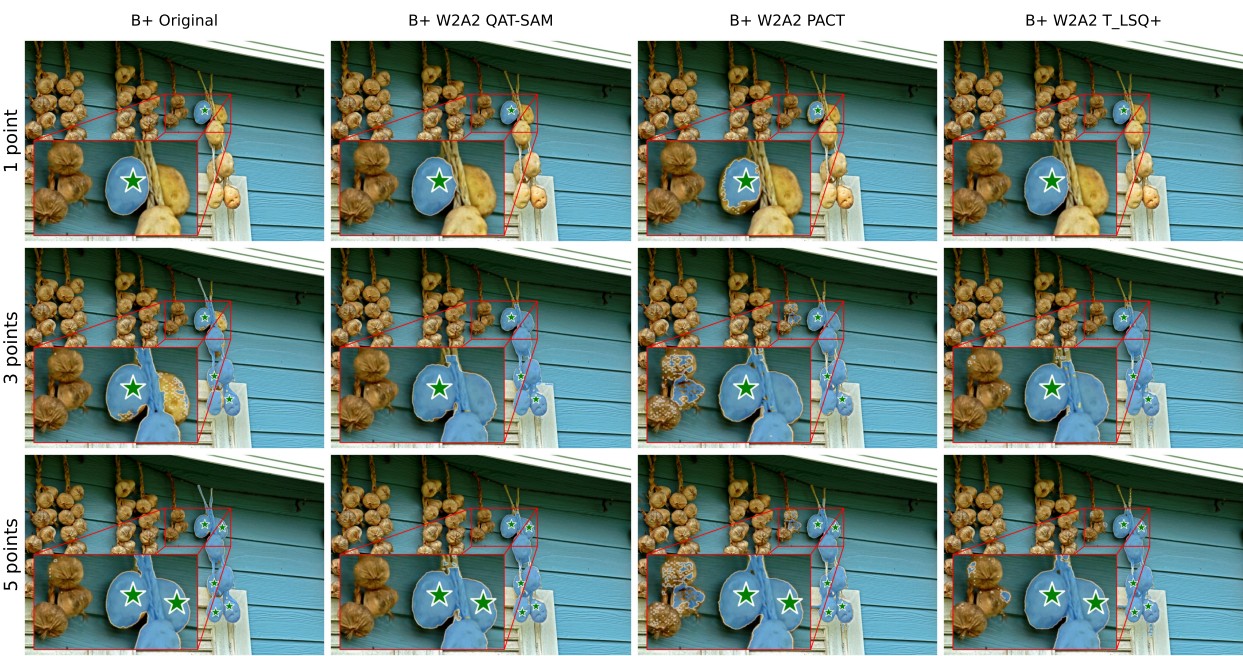

Figure 13: Qualitative results of SAM2, QAT-SAM(ours), PACT, and T_LSQ+ on the base plus architecture. From top to bottom, we add 1, 3, and 5 prompt points.

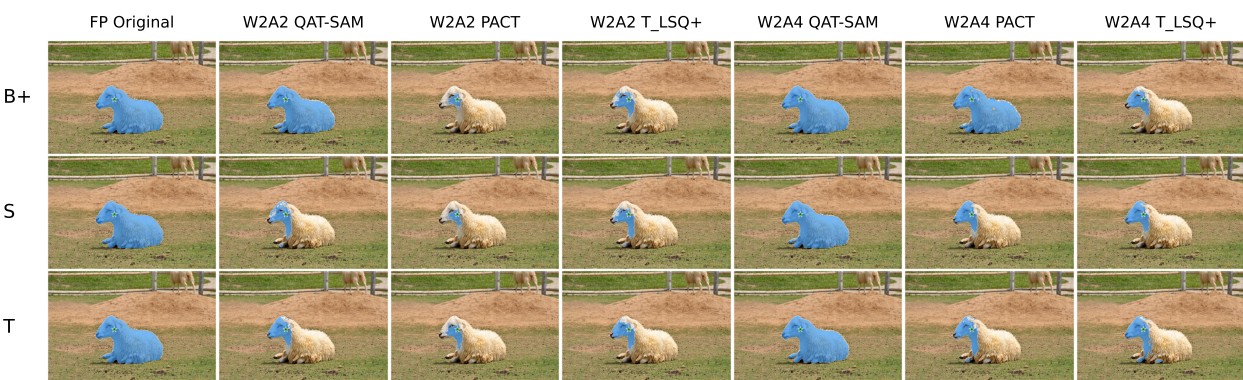

Figure 14: Qualitative results of variants of models with 1 point as prompt.

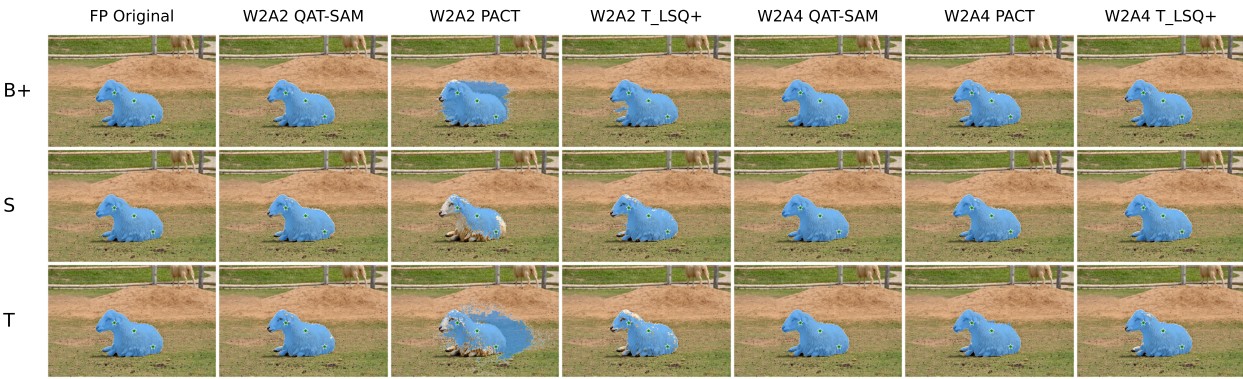

Figure 15: Qualitative results of variants of models with 3 points as prompt.

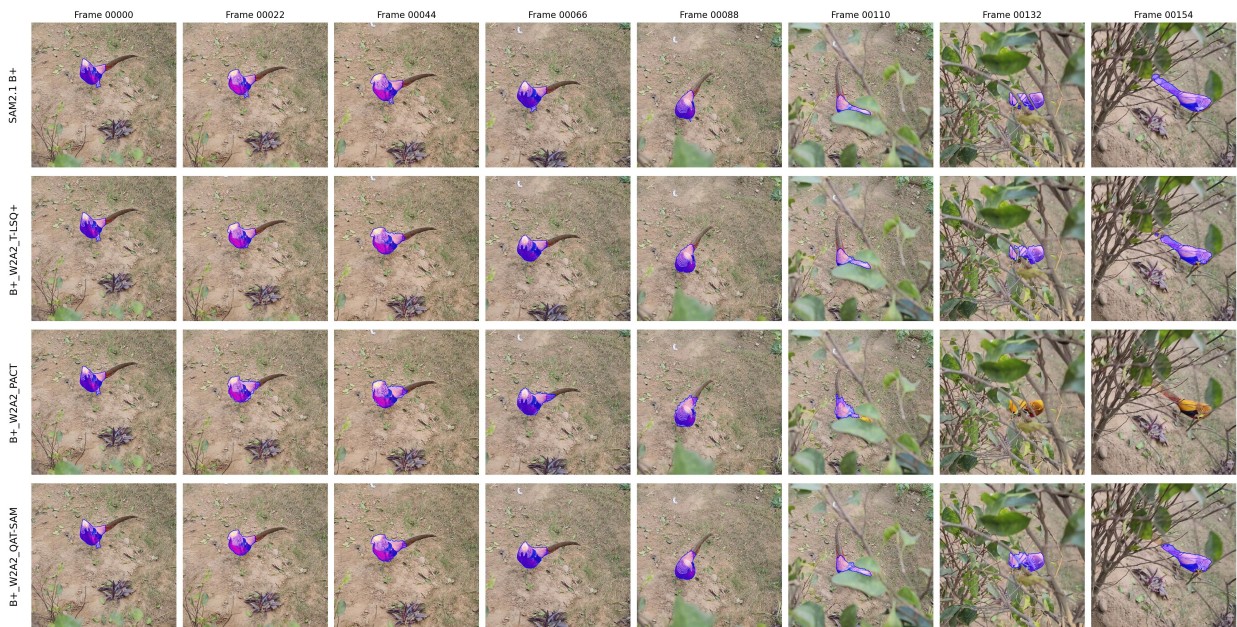

Figure 16: Qualitative results of variants of models SAM2, PACT, T_LSQ+, and QAT-SAM(ours) on the base plus architecture for a video sequence. The first column shows the input mask provided in the first frame, while the remaining columns display the predicted segmentations for each model across subsequent frames.

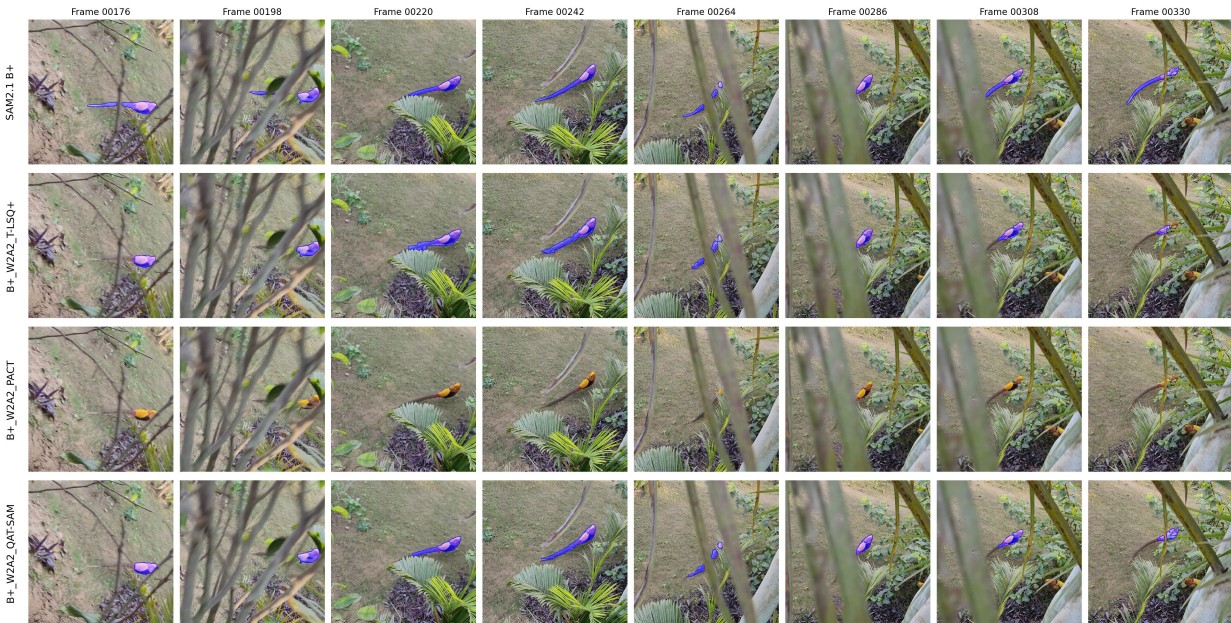

Figure 17: Continuation of qualitative results from Figure 16, showing additional frames from the same video sequence.

