# OpenReview forum: "QAT-SAM: Accurate Quantization for Segment Anything Model 2"
_TMLR — Decision pending for TMLR_

### Review · Reviewer_U99H · 2026-04-20

**Summary Of Contributions:**

This manuscript proposes QAT-SAM, which quantizes SAM2 in ultra-low bits. The first component of VRS reduces weight variance for easier quantization. The second component of LSC applies outlier clipping in weight and activation, applying QAT. Experiments on several datasets confirm consistent improvements.

**Audience:**

Yes

**Audience Explanation:**

Several researchers in specific fields, such as quantization engineers, may find certain values in this study.

**Claims And Evidence:**

No

**Claims Explanation:**

I think the current version is not sufficient. Please see the Requested Changes below.

**Requested Changes:**

- The VRC needs further study and validation. The authors said to use fallback to keep the original one; VRC is interesting but looks data-dependent and layer-dependent. More thorough investigation to make it clear without heuristics would make VRC more convincing.
- What would be the specific insight that is learned from SAM2, rather than general quantization for vision transformers? The proposed method looks specific to SAM2 rather than general quantization to vision transformers, but the underlying difference from SAM2 should be clearly presented.
- The left-hand side of Eq. 2 should be \hat{W}^\top, not Y. The same goes for Eq. 3.
- “since the weight means are generally close to zero” → This should be explicitly confirmed for SAM2.
- If k \in R_+, it would not require an absolute value operation for computing the clipping bound.
- Check typos:
    - “From Equation equation 2”
    - “minimum Frobenius norm weight matrix \hat{Y}”
    - “agorithm”
    - “Section B.1.3 the LSQ+ …” → ‘In Section B.1.3, the LSQ+ …”
    - “insignts”
    - “soulution”
    - “W4TA4” → “W4A4”
    - “in all losses all configurations” → “in all losses for all configurations”
- Check references. I would like to flag AI hallucinations for citations.
    - Incorrect authors for BRECQ.
    - Incorrect authors for “Up or Down? Adaptive Rounding for Post-Training Quantization”
    - The same goes for incorrect authors for PTQ4SAM.
    - Incorrect authors for QuaRot.
    - Incorrect authors for DoReFa-Net. I wonder where the authors extracted the bib entries. DBLP? Google Scholar? or GPT/Claude/Gemini?
    - “Quantized neural networks: training neural networks with low precision weights and activations” should be JMLR 2018, not JMLR 2017.
    - HQQ links are dead. Possibly, it should be dropbox.github.io/hqq_blog, not mobiusml.github.io/hqq_blog.

---

> ### Author Response · Authors · 2026-05-07
>
> Dear Reviewer U99H,
>
> Thank you for your detailed review. We sincerely appreciate you flagging the typos, notation issues, and the bibliography errors.
>
> We have also addressed your major conceptual concerns:
>
> 1. **VRC Heuristics and Data-Dependency:** We completely agree that a single global parameter $\lambda_0$ cannot optimally match the varying alignment between weight mass and activation singular directions across different layers. To address this, we have developed and introduced an **Adaptive VRC (AVRC)** strategy, detailed in a new appendix subsection (A2). Instead of a heuristic global fallback, AVRC operates on a layer-specific basis. For each layer $\ell$, it adaptively sets the parameter $\lambda_\ell$ to the largest value that satisfies a strict, single relative output-deviation budget $\delta$, alongside a variance-reduction constraint. This replaces the heuristic global fallback and makes VRC much more robust and convincing. To directly address the data-dependency concern, we additionally ran AVRC ($\delta = 0.1$) across 10 independently drawn calibration sets on the B+ encoder and observed that both the global per-layer standard-deviation reduction ($17.94\% \pm 0.16\%$) and the downstream COCO mAP ($48.0 \pm 0.1$ ppt) are essentially constant across seeds, indicating that AVRC's per-layer regularizer choice is driven by the layer's spectral structure rather than by the particular calibration images.
>
>
> 2. **Specific Insight for SAM2 vs. General ViTs:** We have added clarification (Appendix C) that SAM2's Hiera encoder is measurably harder to quantize than standard ViT encoders. For example, applying SmoothQuant (a standard ViT technique) to SAM1 (W3A8) loses only 5.2 ppt, whereas on SAM2 it collapses by 48.5 ppt. The two effects we directly measure are (1) heavily-tailed weights (outliers consume disproportionate range) and (2) input-dependent activation shifts within a single layer (which static clipping cannot adapt to, motivating our EMA-tracked LSC). Hiera's hierarchical token-merging structure, in contrast to a uniform-resolution ViT, plausibly amplifies these effects across stages, but our claim in Appendix C rests on the measured statistics rather than on the architectural argument alone. Our method is calibrated against this measured profile.
> 3. **Weight Means Close to Zero:** As requested, we have explicitly confirmed this for SAM2 (with an average value of $-2\times e^{-4}$ for linear layers in the B+ encoder, and a maximum of $-1\times e^{-3}$)
>
> We believe these additions provide the thorough investigation and principled formulation you requested. Thank you again for your valuable feedback.

---

### Review · Reviewer_Sysd · 2026-04-23

**Summary Of Contributions:**

This paper presents QAT-SAM, a quantization-aware training (QAT) scheme for SAM2 (segment anything models). The authors notice that (P1) SAM2 models Heira encoder (hierarchical ViT) has high variance weight outliers in linear layers, as well as (P2) heavy-tail activations with large input-dependent dynamic range shifts within the same layer. They particularly demonstrate this to be a significantly bigger issue for SAM2 compared to SAM and SAM3. To address these they come up with two main contributions:  (1) Variance-reduced calibration (VRC): a pre-QAT initialization scheme to reduce weight variance (P1) which projects the weight matrices for linear layers into the activation-subspace of a few calibration dataset points using an L2-regularized pseudo-inverse. (2) Learnable Statistical Clipping (LSC): QAT algorithm for learnable clipping to reduce activation heavy-tails (P2) that uses EMA statistics along with a single learnable scalar per-layer (per-quantizer) demonstrating more stable training gradients compared to baselines (LSQ+).

Across main paper and appendix, authors provide comprehensive evaluation across 3 different encoder sizes, 2 different quantization configs, various datasets and benchmarks and also compare against both QAT and PTQ (post-training quantization techniques) showing significant improvement over the baselines wrt the Pareto-optimal frontier for (performance, model) size tradeoff. The provided visual results are also compelling.

As explained further below, the main weakness for this work in my opinion is the theoretical motivation and justification for both the methods, in particular VRC along with demonstrated gains.

**Additional Comments:**

N.A.

**Audience:**

Yes

**Audience Explanation:**

In my opinion, this is a good technical report. The techniques might be relevant for other architectures or models which use similar Hiera backbones, or for practitioners interested in deploying SAM2 in future quantized hardware kernels.

**Broader Impact Concerns:**

N.A.

**Claims And Evidence:**

Yes

**Claims Explanation:**

Authors provide a comprehensive evaluation of their techniques compared with the baselines and demonstrate improvements over the baselines.

**Requested Changes:**

## Strengths
- Comprehensive evaluation as described above.
- Compelling numerical results demonstrating techniques providing significantly better Pareto-frontier.
- Simplicity of both proposed stechniques.

## Weaknesses, questions and requested changes
 Though the work has compelling empirical evidence, I am not convinced with the presentation and claims regarding the origin of techniques and why they work. The main concern is around VRC which shows minimal gains compared to LSC in the ablation experiments presented by authors.

1. The paper's motivation for using pseudo-inverse for VRC is to reduce Frobenius norm of the weights while keeping the activation norm close in L2 sense. The authors are essentially solving $\\|XW^T - X\hat{W}^T\\|^2 + \lambda \\|\hat{W}\\|^2$ where $W$, $\hat{W}$ are the original and calibrated weights, $X$ are the inputs, $XW^T$ the activations over the calibration set, and justify the use of Moore-Penrose pseudo-inverse by saying it minimizes the Frobenius norm and hence weight matrix variance. Specifically in Section 3.2 they say, "The use of the Moore-Penrose pseudoinverse is motivated by its known properties in terms of 1) minimizing the residual with the calibration output activations and 2) minimizing the Frobenius norm of the obtained weights." However one needs to note that the pseudo-inverse has different interpretations for overdetermined vs underdetermined set of equations. For the overdetermined case $(N > d)$, point 2 is vacuously true as we have a unique minimizer for $\hat{W}$ which is automatically minimum-norm, whereas for the underdetermined case $(N < d)$ we have many minimizers which give the error exactly 0 but the pseudo-inverse solution minimizes the Frobenius norm of $\hat{W}$. In this case, my guess is we are mostly working in an overdetermined regime for most layers $(N > d)$ and hence the solution we are trying to find is the unique least-squares solution, not a non-trivial minimal Frobenius norm solution. I think the main reason the scheme is working is because of the regularization term which reduces the norm of weights and hence the outlier behavior in weights. This is also the probable cause why authors had to tune a $\sigma_*$. Authors probably also see varying advantages across layers in Fig 6 because of the same reason.
    1. I think authors should clarify it in the text and clarify the mechanism using which the pseudo-inverse is actually helping the overall process. The main reason VRC is able to solve P1 is because of regularization. Whether VRC reduces extreme max outliers (Appendix C) is not guaranteed and depends on whether outlier weight values lie in low-activation eigendirections.
    2. Similarly Fig 5b is misleading as we are comparing quantization error at a given $\lambda$. This quantity will trivially decrease with $\lambda$ as the weights will become smaller because of the L2 norm. We should look at relative error (e.g., normalized by the quantization error of the original weights, or the output reconstruction error after quantization).
    3. The authors should also have a comparison with MagR (Zhang et al. 2024) by replacing VRC in the current experiment and instead using MagR for the preconditioning before QAT. The reason is MagR has a very similar objective but it tries to minimize the $L_\infty$ norm instead of L2/Frobenius norm, and it is not clear to me which should work better in conjunction with QAT techniques.
    4. Finally, the presentation in equations 2–3 and around them is very unclear with confusing notations. $Y$ is used both for activations and for the calibrated weights ($\hat{Y}$). Similarly $p$ and $d_\text{in}$ are used interchangeably. Authors should use consistent and clear notations.
    5. Similarly, the stated objective for VRC as written (and presented in Fig 2) is vacuous as $\arg\min \text{Var}(W)$ without the output constraint gives $W=0$. The constraint that activations remain similar is essential and should be stated explicitly in the main text alongside the objective.
    6. The code is missing both the VRC calibration script and the `calib_weights_lambda2.pt` checkpoint which doesn't allow its reproduction.
2. Similarly, it is unclear to me how LSC is statistically motivated? I agree it works and is simpler than LSQ+ but to claim it is statistically motivated seems to be a stretch. The EMA tracking of $(\mu, \sigma)$ is a reasonable statistical choice (analogous to BatchNorm estimators), but the specific clipping rule $\mu \pm k\sigma$ implicitly assumes a symmetric distribution around the mean where $k\sigma$ is the right measure of spread. Neither of these is verified for SAM2's heavy-tailed activation distributions that the paper itself identifies as the central challenge in P2.

### Minor:

Suggestions:

- Maybe worth clarifying in Table 1 that the FP numbers are quoted from official release of SAM2 (as described in Appendix B.1.2) so that a reader notes that the fall in performance is also because of training pipeline limitation beyond just an effect of quantization.
- Would be nice to see how Figure 3 caption numbers (10-20% reduction) and Figure 6 caption numbers (max 38.7% and mean 13.2%) are related.

Typos:

- Section 3.2, P5: “calibration “agorithm” -> calibration algorithm”
- Section 3.1, P4: wrong appendix citation “Appendix B.1.2” -> “Appendix C” (B.1.2 covers training pipeline differences, not quantization comparisons)
- Section 4.2, P9: “x8 larger” -> “8x larger”
- Figure 16, P27: caption has 3 but figure has 4 methods

---

> ### Author Response · Authors · 2026-05-07
>
> Dear Reviewer Sysd,
>
> Thank you for your very careful read of our paper and the insightful feedback. We have addressed the typos and minor corrections you pointed out, and we have made significant revisions to clarify the mechanisms of our proposed methods:
>
> 1. **VRC Mechanism, and Variance vs. Outliers:** Thank you for the careful read. The regime distinction you draw is important and we want to be precise about how it interacts with our design. The pseudoinverse formulation in Eq. 2 is itself regime-agnostic: with very few calibration images the system is underdetermined and the formulation selects the genuine minimum-Frobenius-norm solution, while with many images it reduces to the unique least-squares solution. With our default $n{=}50$-image calibration the relevant layers fall into the overdetermined regime ($B \gg d_{\text{in}}$), so the "minimum Frobenius norm" property is vacuous in that specific setting; we agree that, there, the active mechanism for variance reduction is the Tikhonov term in Eq. 3, the per-layer variation in Figure 6 reflects the alignment you describe between weight mass and low-activation singular directions, and the role of $\lambda$ is to trade output deviation against shrinkage strength. Section 3.2 has been rewritten to make this regime-specific reading explicit.
>
>     On rereading Section 3.2 in light of your comments, we noticed that our original wording may not have made one closely related distinction clear enough, and we would like to take the opportunity to address it here: in our pipeline, VRC is intended as a *variance-reduction* step rather than an outlier-reduction step. The two-stage design described in Section 3.1 (and restated in Sections 1 and 5) keeps these roles separate: VRC reduces weight variance via Frobenius-norm regularization, while LSC then clips the residual extreme values during QAT. Since reducing the per-channel standard deviation and reducing the maximum absolute value are mathematically distinct objectives, we wanted to underline this division of labor explicitly here.
> 2. **Comparison with MagR:** Per your suggestion, we benchmarked MagR (Zhang et al., 2024) as an alternative preconditioner before our QAT/LSC stage. MagR's $L_\infty$-norm minimization explicitly targets extreme values, so plugging it in upstream of LSC is a natural test of whether outlier handling is the appropriate job for the calibration step. In practice, MagR + QAT-SAM (LSC) reaches 37.6 mAP versus QAT-SAM's 39.2 (for B+ W2A2 mAP). The 1.6 ppt gap is consistent with the division of labor described in point 1: VRC (variance) and LSC (extreme-value clipping) target different quantities and compose well, whereas MagR and LSC both target extreme values and partially overlap.
> 3. **Statistical Motivation of LSC and Symmetry:** We do not claim Gaussianity or that $\sigma$ is the optimal scale parameter for heavy-tailed data. The "statistical" label refers to using sample moments (EMA) rather than free parameters as the scale, analogous to BatchNorm's running estimators. Regarding symmetry: the bound $[\mu-k\sigma, \mu+k\sigma]$ is mapped via the affine zero-point to the integer grid, so the grid shifts to follow the data; we do not assume the distribution itself is symmetric around zero, only that the clip range is symmetric around the mean. We did explore the asymmetric extension with two scalars ($k_{\text{left}}, k_{\text{right}}$) per quantizer during the initial design phase, but the symmetric version consistently performed better in our experiments, so we adopted it as the final design.
> 4. **Reproducibility:** The new supplementary material includes the AVRC procedure (Appendix A.2), in the previous one we released the calibrated checkpoint for the purpose of complete reproduction.
> 5. **Figure 5b normalization:** You are right that comparing absolute quantization MSE at a fixed $\lambda$ is misleading, since the calibrated weights have smaller $L_2$ norm by construction and so absolute MSE shrinks trivially. We plan, for the final release of the paper, to replace Figure 5b with a normalized version reporting the ratio of $\hat{\mathbf{W}}_\lambda$'s quantization MSE to that of $\mathbf{W}$ (and, on the same plot, the post-quantization output reconstruction error), so the figure exposes relative improvement rather than the trivially decreasing absolute quantity.
> 6. **Minor Points:** We clarified in Table 1 that the FP numbers are from the official SAM2 release, and we thank you for catching the notation inconsistencies in Equations 2-3, which have been corrected.
>
> We appreciate you pushing us for these clarifications, which have significantly strengthened the paper's theoretical and empirical rigor.

---

> > ### Comment · Reviewer_Sysd · 2026-05-26
> >
> > Thanks for the responses and addressing them.
> > - I would still recommend adding MagR finding in the main paper e.g. Table 2.
> > - I would also recommend Fig 5b normalization to be fixed before final publication.

---

> > > ### Author Response · Authors · 2026-06-04
> > >
> > > Dear Reviewer Sysd,
> > >
> > > Thank you; we will make sure to include your suggestions in the paper.

---

### Review · Reviewer_coY5 · 2026-04-24

**Summary Of Contributions:**

This paper proposes QAT-SAM, a quantization-aware training pipeline for SAM2, built from two components: VRC (a regularized pseudoinverse-based calibration step applied to encoder linear layers) and LSC (a learnable clipping scheme based on EMA-tracked mean/std with one learnable scalar per quantizer). The paper evaluates the method on SAM2.1 tiny/small/base-plus models for VOS (SA-V, MOSE) and instance segmentation (COCO2017), mainly in W2A4 and W2A2 settings. Empirically, the method improves clearly over the chosen QAT baselines.

**Additional Comments:**

**Strengths:**
*   **Methodological Elegance:** The LSC method is intuitive and tackles a well-known flaw in aggressive low-bit QAT (the instability of learning thousands of per-channel scales/zero-points from scratch). Grounding the bounds in EMA-tracked moments while restricting the gradient-based search to a single parameter ($k$) is a very sound approach.
*   **Clear Component Ablation:** The ablation studies (Tables 2, 3, and 9) do an excellent job isolating the independent benefits of VRC and LSC, proving both components contribute meaningfully to the final performance.
*   **Well-Written and Clear:** The paper is logically structured, the visualisations (e.g., Figure 5 on bias-variance tradeoff in VRC, and the qualitative masks) are helpful, and the structural challenges of quantizing SAM2 are clearly articulated.

**Weaknesses:**

1.  **Outdated Baselines and Overstated SOTA Claims:** For a 2026 submission, the comparison set is notably outdated. The main QAT baselines (PACT, LSQ+, MinMax) originate from 2018-2020. Relying on methods proposed over five years ago is insufficient to substantiate the claim of achieving a "new state-of-the-art QAT." Furthermore, to get LSQ+ to work, it is weakened into an activation-only variant (T_LSQ+) with MinMax weights because the original failed to converge. While this is a practical engineering fix, it significantly weakens the SOTA claims, especially given the lack of comparison against more recent (2022-2026) transformer-specific quantization methods.
2.  **Incomparable PTQ Baselines (Not Apples-to-Apples):** The comparison in Figure 1 and Table 7 between QAT-SAM and PTQ methods is flawed. The authors compare their quantized SAM2 (Hiera architecture) against PTQ results reported for the original SAM1 (ViT-B/L/H architectures). Since the backbone and model families fundamentally differ, the claimed Pareto frontier is misleading. This does not provide valid evidence that QAT-SAM is superior to standard PTQ on the *same* architecture.
3.  **Incomplete Efficiency and Deployment Story:** The paper is heavily motivated by the need for deployment on resource-constrained edge devices. However, Section 4.2 explicitly notes that there are no deployable W2 kernels available, meaning the paper only reports theoretical BitOPs rather than real inference speedups. Furthermore, the proposed pipeline is exceptionally resource-intensive: the calibration step ($n=50$) requires 263.7s, 81GB RAM, and 29.9GB VRAM on an H100, and the training requires 8x A100 GPUs for up to 35 hours. This steep computational overhead contradicts the practical, edge-friendly narrative.
4.  **Overstated "High Fidelity" Claims:** The framing of the method as achieving "high fidelity" is not fully supported by the absolute numbers. For example, on the SAM2.1-B+ model, the SA-V validation score drops from 78.1 J&F in FP16 to 64.7 (W2A4) and 55.2 (W2A2). Similarly, COCO mAP drops from 49.6 to 45.8 (W2A4) and 39.2 (W2A2). While the relative gains over the weak baselines are real, an absolute drop of ~23 points in J&F does not constitute "high fidelity."
5.  **Scope of Application vs. Method Generality:** Both VRC and LSC are fundamentally architecture-agnostic techniques, yet they are tested *exclusively* on SAM2. Without results on standard Vision Transformers (e.g., ImageNet classification) or LLMs, it is difficult to determine if this is a general breakthrough in ultra-low-bit QAT or an approach overfit to SAM2's specific distribution quirks.

**Audience:**

Yes

**Audience Explanation:**

Yes. Researchers and practitioners working on efficient deep learning, model quantization, and the deployment of vision foundation models will find this paper relevant.

**Broader Impact Concerns:**

There are no specific, immediate ethical concerns introduced by this paper that require a dedicated Broader Impact statement beyond standard practices.

**Claims And Evidence:**

Yes

**Claims Explanation:**

Yes. The empirical evidence on SAM2 supports the claims that VRC and LSC improve 2-bit and 4-bit QAT performance.

**Requested Changes:**

1.  **Revise SOTA and "High Fidelity" Claims:** Tone down the claims regarding "state-of-the-art" performance and "high fidelity." Acknowledge the large absolute accuracy degradation in ultra-low bit-widths. The framing should focus on *improving robustness over standard/historical baselines at extreme bit-widths* rather than claiming a solved, high-fidelity Pareto frontier.
2.  **Fix the PTQ Comparison:** Either provide an apples-to-apples comparison by running a standard PTQ method (like SmoothQuant, PTQ4SAM, or GPTQ/OmniQuant variants) directly on SAM2, or heavily caveat Table 7 and Figure 1 to explicitly state that the architectures are different and the comparison is only illustrative of general model size vs. performance trends across different SAM generations.
3.  **Contextualize Baselines and Add Modern Comparisons:** Explicitly acknowledge the age of PACT/LSQ+/MinMax in the text. Ideally, include at least one comparison to a more recent (post-2022) quantization method designed for ViTs or Foundation Models. At a minimum, justify why these specific older baselines were chosen and detail how T_LSQ+ differs from a modern SOTA baseline.
4.  **Clarify the Deployment Paradox:** Add a paragraph in the discussion or conclusion addressing the practical limitations of the method. Explicitly note the lack of current W2 hardware support and discuss the high computational cost of the VRC calibration and QAT training stages relative to the edge-deployment motivation.

---

> ### Author Response · Authors · 2026-05-07
> **Changes and Comments**
>
> Dear Reviewer coY5,
>
> Thank you for your thorough review and for helping us refine the positioning of our work. We have addressed your requested changes in the revised manuscript:
>
> 1. **Revised SOTA Claims:** We have toned down the claims regarding "state-of-the-art" performance. Throughout the text, we now focus on the framing you suggested: improving robustness over standard and historical baselines at extreme bit-widths, and we explicitly acknowledge the absolute accuracy degradation that still remains in ultra-low bit-widths.
> 2. **Contextualize Baselines and Add Modern Comparisons:** We acknowledge in the text that PACT, LSQ+, and MinMax are older methods, but they remain the de-facto QAT baselines used by prior SAM/ViT quantization works; even recent frameworks such as ParetoQ (Liu et al., 2025c) still build on LSQ, indicating that these baselines are dated but not fully superseded. To address the need for a more modern reference, we have additionally evaluated **StatsQ** (Liu et al., 2023) and included its results in our main table (`Table 1`). StatsQ targets a known QAT pathology in vision transformers called weight oscillation, where in low-bit regimes individual weights flip between adjacent quantization grid points during training and harm convergence; it addresses this by deriving the per-tensor scale from running statistics of the weight distribution rather than learning it as a free parameter, which stabilizes training at low bit-widths and makes it a representative modern baseline for ViT-style backbones. As discussed in Section 4.2, while StatsQ improves upon older baselines like PACT in the W2A4 configurations, our QAT-SAM remains more robust, particularly in the extreme W2A2 setting.
> 3. **PTQ Comparison Caveat:** We have added an explicit caveat in Section 4.2 and in the caption of Figure 1, making it clear that the PTQ baselines are reported on the original SAM (ViT) and that the comparison across architectures is only illustrative of general model size versus performance trends.
> 4. **The Deployment Paradox:** We agree with your point regarding practical limitations. We have added a dedicated Limitations section (Section 5) that explicitly notes the lack of current W2 hardware support for edge deployment and acknowledges the non-trivial computational cost of the VRC calibration and QAT training stages.
>
> Thank you again for your constructive feedback and for catching the minor typos, which have been corrected.